

# Spatio-temporal distribution of ostracod species in saline inland lakes (Mansfeld lake area; Central Germany)

Marlene Hoehle[1] and Claudia Wrozyna[2]

[1] Institute for Geophysics and Geology, Universität Leipzig, Leipzig, Germany
[2] Institute for Geography and Geology, Universität Greifswald, Greifswald, Germany

Corresponding authors
Marlene Hoehle,
marlene.hoehle@uni-leipzig.de
Claudia Wrozyna,
claudia.wrozyna@uni-greifswald.de

## ABSTRACT

Ostracods are a diverse group of microcrustaceans with a ubiquitous distribution in a wide array of aquatic habitats and are common constituents of lake sediments. Inferences on the temporal-spatial distribution of ostracod species is a prerequisite for reconstructions of palaeoenvironmental conditions. This requires a precise knowledge not only about ecological preferences and specific life histories, but also the understanding how (local) ecological parameters affect ostracod species assemblages (abundance and composition). Generally, these studies are rare and often characterized by an insufficient differentiation of living specimens from the total amount of valves of the modern population leading to uncertainties in species occurrences and diversity data. Modern ostracod populations were sampled from 12 water bodies within a relatively small study area (Mansfeld lake area, Central Germany). Physico-chemical parameters (temperature, oxygen content, conductivity, pH) were measured *in situ* and the uppermost 2 cm of sediment were collected in different seasons (April, June, September). Relative abundances of ostracods (living and dead), differentiated for adults and juveniles, were used for statistical analyses (Spearman's rank correlation, Canonical correspondence analysis, Cluster analyses, Fisher's $\alpha$), to investigate relationships between species distribution and environmental factors as well as to identify habitat similarities and ostracod species assemblages. In total, 27 ostracod species (20 living species) were identified. Majority of them are considered as very common (cosmopolitan) freshwater species. Only two species are usually known from brackish water (*Cytheromorpha fuscata* and *Cyprideis torosa*). This is the first confirmation of living *C. torosa* in German inland waters. The relative abundances of ostracods show strong fluctuations during the study period and differences in composition of the ostracod species assemblages between and within the water bodies. There are also strong differences between bio- and taphocoenoses. The measured physico-chemical parameters which are usually considered as most important drivers on ostracod species distribution do not contribute to explain the observed temporal-spatial distribution of the ostracod species. Differences in taphocoenoses show, that taphonomic processes can be very local and the sampling site, as well as the sampling time, is crucial. Biodiversity of ostracods is biased by sampling time, the variability of the ostracod assemblages between sampling month and the relationship between abundance of valves and living ostracods is not straightforward. Therefore, without precise knowledge of the ecological requirements of a species at a local scale, uncertainties may exist for the palaeoecological indication of a species.

## INTRODUCTION

Despite their high diversity, vast array of ecosystem services, basic science on invertebrates is scarce and underfunded which contributes to that most species remain undescribed. Scarce population time series of invertebrates contribute to poor knowledge of the spatial and temporal distribution of species as well as their ecological preferences is mostly unknown (*Baillie et al., 2008*; *Cardoso et al., 2011*; *Outhwaite et al., 2020*).

Ostracods (bivalved microcrustaceans), are one of the most diverse groups of living benthic invertebrates (*Horne, Meisch & Martens, 2019*); they inhabit almost all types of aquatic environments: marine, freshwater and even some semi-terrestrial (*Horne, Cohen & Martens, 2002*). In particular, freshwater ostracods are of great interest in a variety of ecological and evolutionary studies because they are partially found *en masse* in aquatic environments and their calcified valves are common and well preserved in lake sediments (*Martens et al., 2008*) which enable to study long-term trends *e.g.*, biodiversity. Although ostracods have generally very broad ecological tolerances, distribution is mainly determined by a variety of physico-chemical (abiotic) factors (temperature, pH, salinity, oxygen content, substrate type and depth) (*Mezquita et al., 2005*), as well as biotic factors (competition, predation, commensalism) (*Guisan, Thuiller & Zimmermann, 2010*) and each species has specific tolerances and preferences to these factors (*Kiss, 2007*). Especially, temperature and salinity are considered as major controlling factors for distribution of freshwater ostracod species (*Ruiz et al., 2013*).

Precise autecological inferences based on fossil material depends on extensive knowledge about the ecological requirements and life-cycle information of the different species, as well as additionally (local) influences on these. Nonetheless, comprehensive studies on these are still rare (*Kulkoyluoglu, Yilmaz & Yavuzatmaca, 2017*; *Viehberg, 2005*). However, life cycle (or seasonal distribution), ecology, and transfer into the geologic record (taphonomic processes) have rarely been considered together. This, in turn, is problematic, as sometimes incorrect diversity estimates are obtained and, most importantly, imprecise assumptions are made about ecological preferences or (subtle) responses to environmental changes are overlooked, which leads to a discrepancy between recent and fossil data. Also problematic is the often insufficient differentiation between living ostracods (biocoenoses) and valves (taphocoenoses). Diversity and autecology of species are usually evaluated on the basis of recent material (but usually only valves). Species or entire assemblages may be, thus, assigned to incorrect environmental conditions. In addition, studies are mostly based on large-scale trends, which are transferred to local scales (*e.g.*, *Pint et al., 2017*).

The Mansfeld lake area provides within a narrow geographical area several water bodies differing slightly with respect to hydrochemistry, hydro(geo)logy or degree of pollution settled. These water bodies offer as 'natural laboratories' ideal conditions to study the effects of regional and local environmental parameters and their seasonal fluctuations on the distribution of individual ostracod species, but also on the entire community.

The main objective of this study is to characterize ostracod species in slightly saline inland waters and their spatio-temporal distribution. For this purpose, the following research questions were defined: (1) Are there disparities between the water bodies in terms of physico-chemical characteristics and ostracod assemblages? (2) By repeated sampling in three months, can differences in species abundance and richness of living ostracods be determined, which may provide inferences about the species life cycle? (3) Can conclusions on taphonomic processes in the waterbodies be drawn, by differentiating between living ostracods and valves? (4) Are possible variations in living ostracod assemblages (abundance and richness) related to the physico-chemical parameters of the water bodies?

## MATERIALS AND METHODS

### Study area

The Mansfeld lake area is located 25 km west of Halle (Saale), a city of Saxony-Anhalt, in the centre of the central German dry region (Fig. 1). It is characterized by low precipitation (428 mm/a), a negative water balance, as well as short and pronounced extreme rainfall events (*Wennrich, 2005*). The regional climate is characterised by a warm-toned mesoclimate, with subcontinental tendencies and an annual mean temperature of 8.8 °C (*Trost & Rauchhaus, 2000*; *Wennrich et al., 2005*). Another peculiarity of the region is the natural salinity of the water bodies, which is caused by saline inflows of salt deposits of Permian age from the underground (*Wennrich, Meng & Schmiedl, 2007*). Most of the investigated water bodies (KS, BS, TE, TE2, QW and MG) lie within the area of the former lake 'Salziger See' (overview of the sampling localities and their abbreviations are provided in Table 1). North of the former Salziger See is 'Süßer See', which is directly connected by ditches with KS, BS and MG. Both (Süßer See and former Salziger See) are located in the deepest part of a large depression (Teutschenthal Anticline) (*Wennrich, Meng & Schmiedl, 2007*). The remaining water bodies (OT, OT2, WL, TA, ST) are located outside this area, topographically higher and with a maximum distance of 3 km, but mostly located at the marginal area of the former Salziger See (Fig. 1). Although the water bodies presumably all have the same catchment, they have different sub-catchments, and the water bodies outside the anticline have different hydrological conditions (*e.g.*, saline inflows) (*John et al., 2000*).

### Sampling and data collection

Three sampling campaigns were carried out in 2019 (April 16, June 7 and September 12) in the Mansfeld lake area. During each sampling campaign, 10–12 water bodies (ponds, small lakes, ditch) were investigated. An overview about the sampling localities is provided in Table 1.

Generally, we tried to visit the same sampling site at each water body during each campaign as far as possible. Exceptions were made at the lakes TE and ST due to limited access due to dense vegetation of the first sampling site. The sampling localities differed ∼80 m north (TE) and 120 m east (ST), respectively, from the first site.

Physico-chemical variables (water temperature, electrical conductivity, pH and oxygen content) were measured *in situ* using probes of the company WTW. Salinity was measured *in situ* as electrical conductivity and converted (*Rice, Baird & Eaton, 2017*) to salinity for

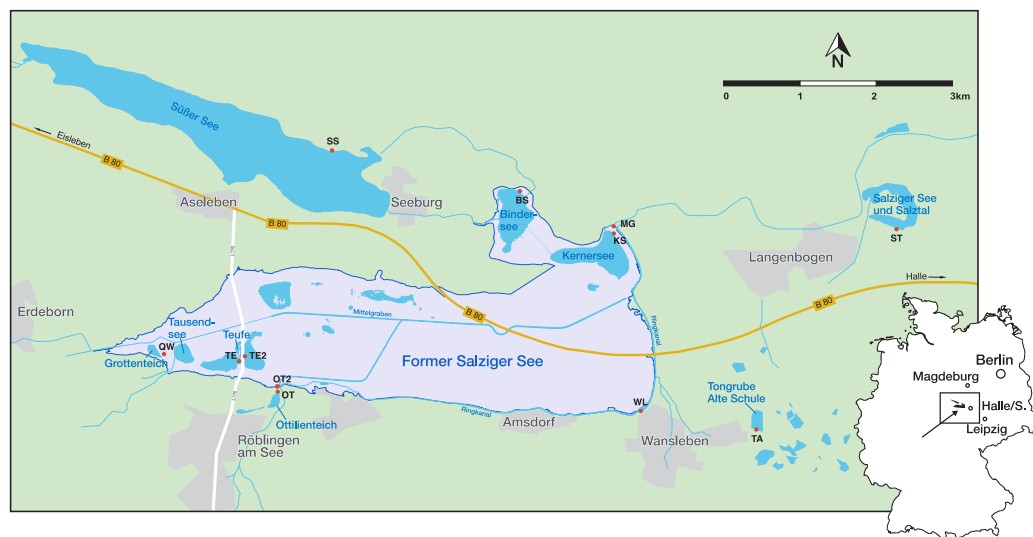

**Figure 1** **Geographical location in Germany and sampling sites (red points) in the Mansfeld lake area. Outlines of the former Salziger See are taken from (*Trost & Rauchhaus, 2000*).** Abbreviations of the water bodies: KS, Kernersee; BS, Bindersee; TE, Teufe; TE2, Teufe2; QW, Quelle im Wald; OT, Ottilienteich; OT2, Ottilienteich2; WL, Wannsleben; TA, Tongrube Alte Schule; ST, Salzatal; SS, Süßer See, MG, Mittelgraben.

**Table 1** **Overview of sampling sites in terms of name and water body abbreviations, coordinates, habitat type and sample month.** Sample month: A = April, J = June, S = September.

| Name | Abbreviation | Latitude (N) | Longitude (E) | Habitat type | Sample month |
|---|---|---|---|---|---|
| Kernersee | KS | 51.485102 | 11.740385 | open shoreline of a lake, sand | A/J/S |
| Bindersee | BS | 51.489583 | 11.724183 | reed-belt of the lake, sand with detritus | A/J/S |
| Teufe | TE | 51.470694 | 11.675120 | die-back reed-belt of the lake, muddy substrate (sludge)with other plant remains | A/J/S |
| Teufe 2 | TE2 | 51.471072 | 11.676226 | reed-belt of the lake, mud (sludge) | A/J |
| Quelle im Wald (spring in the woods) | QW | 51.470166 | 11.668428 | temporary source fed ditch section, mud with plant remains (rotten leaves) | A/J/S |
| Ottilienteich | OT | 51.467471 | 11.682365 | braced shoreline of the lake, dense growth of aqutic macrophytes, rarely sediment | A/J/S |
| Ottilienteich 2 | OT2 | 51.468329 | 11.682151 | open shoreline of the lake, sand | A/J/S |
| Wannsleben | WL | 51.465623 | 11.744928 | shallow shoreline of the pond, muddy substrate with gravel | A/J/S |
| Tongrube Alte Schule (Clay pit 'Old School') | TA | 51.463703 | 11.764960 | open shoreline, muddy substrate with algea and macrophyts | A/J/S |
| Salzatal | ST | 51.485483 | 11.790107 | reed-belt of the lake, sand | A/J |
| Süßer See | SS | 51.498387 | 11.676360 | shallow shoreline, sand covered with coarse gravel, stones and algea | J/S |
| Mittelgraben | MG | 51.485583 | 11.741115 | ditch sampled on ist slope, muddy substrate with macrophytes | S |

direct comparison with literature. All salinity values are in practical salinity units. Sediment samples were collected from the uppermost few (approx.1-2) centimetres of sediment in the littoral zone using a hand net within an area of ∼1–2 m².

The samples were fixed with 70% ethanol in the field at the time of collection, to preserve living specimens. The sediment samples were washed with tap water through standard sieves sizes (1,000 μm, 500 μm, 250 μm and 125 μm). The sieve residues from mesh sizes 500 μm and 250 μm were transferred to sample bags using 70% ethanol, mesh size 125 μm were oven-dried. For some water bodies, 125 μm fraction was considered on a trial basis, but not included in the study due to the extremely small numbers of valves. Ostracods were separated from sediment and species were sorted under a binocular microscope. Ostracods were assumed to have been alive (biocoenoses) at the time of collection, when carapaces were slightly open and with intact and well preserved soft parts. Valves and carapaces (empty or with soft part fragments) found, may include not only dead individuals, but also those shed during moulting (juveniles) and are assumed as taphocoenoses. Due to the fact it is a natural assemblage, taphocoenoses is probably a mix of thanatocoenoses and taphocoenoses (*Boomer, Horne & Slipper, 2003*).

Biocoenoses were stored in ethanol, taphocoenoses were dried at room temperature and stored in micro cells. Ostracods (carapaces, valves, and 'living') were sorted by juveniles and adults. Different juvenile stages were not distinguished.

Species identification was based in most cases on valve morphology according to *Meisch (2000)*, *Fuhrmann (2012)*, and *Wennrich (2005)*. Soft part analysis was performed in cases were species could not be identified from valves only (*e.g., F. fabaeformis, F. holzkampfi, H. incongruens* and *I. gibba*). All ostracod valves were counted (carapace = two valves) and relative abundances (percentage of the sum of valves in each sample) were calculated. Counted ostracods from the two fractions (500 μm and 250 μm) were added together.

## Data analyses

Spearman's rank correlation test was applied to identify relation among relative abundances of living assemblages and physico-chemical parameters.

Canonical correspondence analysis (CCA) was used to investigate the relationships between the distribution of living ostracod assemblages and physical and chemical environmental parameters. Relative abundances of species and four environmental variables including water temperature, electrical conductivity, oxygen content and pH, were used for the CCA.

To calculate Fisher's $\alpha$ diversity and richness of the ostracod species in the different water bodies, the absolute numbers of taphocoenoses of all three months were summed for each species. Additionally, to identify specific ostracod assemblages and determine habitat similarities cluster analyses were carried out.

Cluster analyses were performed based on relative abundances of the species, separately for biocoenoses and taphocoenoses and for water bodies and species. As clustering algorithm, Ward's method to a Euclidian distance matrix was used. In all analyses, samples (water bodies) containing less than 100 individuals were excluded. Furthermore, all species
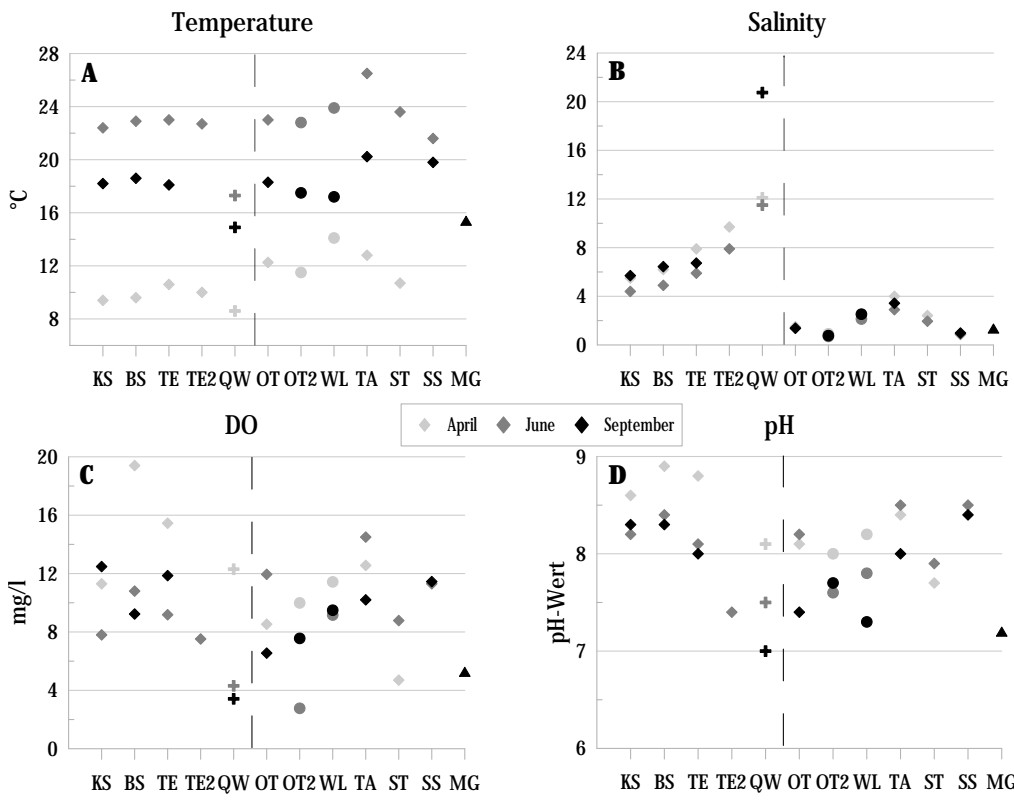

**Figure 2 Physico-chemical parameters of the investigated water bodies in the Mansfeld lake area in terms of temperature, salinity, DO (dissolved oxygen) and pH.** Symbols represent: ♦ = lake, + = temporary source fed ditch section, ● = pond, ▲ = ditch. Dashed lines separate water bodies of the former Salziger See (left) and water bodies outside (right). Abbreviations of the water bodies: KS, Kernersee; BS, Bindersee; TE, Teufe; TE2, Teufe2; QW, Quelle im Wald; OT, Ottilienteich; OT2, Ottilienteich2; WL, Wannsleben; TA, Tongrube Alte Schule; ST, Salzatal; SS, Süßer See; MG, Mittelgraben.

with overall abundances ≤5% and occurring only in one site were also excluded from the analyses.

All statistical analyses were conducted using PAST version 3.25 (2019) (*Hammer, Harper & Ryan, 2001*).

# RESULTS

## Physico-chemical variables and habitat characteristics

Water temperature (ranging from 8.6 °C to 26.5 °C) shows the strongest seasonal fluctuations (highest values in June, lowest in April) (Fig. 2A). The lowest temperatures are provided by the spring (QW). Salinity values (ranging from 0.7 to 20.75) of the former lake Salziger See (KS, BS, TE, TE2 and QW) provide generally higher values than localities outside this area (highest values in September, lowest in April) (Fig. 2B).

Dissolved oxygen (DO) concentrations (ranging from 2.77 mg/l to 19.4 mg/l are in April in most localities higher than in the following months (Fig. 2C).

In all sites, pH values were ≥ 7 at each sampling site and reach a maximum of 8.9 in BS (Fig. 2D). The lowest value of 7 was measured in the spring QW in September.

Substrate texture is dominated by sandy (*e.g.*, BS and KS) and muddy substrate (*e.g.*, TE, TA). Some localities are characterised by algae (SS), detritus (*e.g.*, BS) or aquatic macrophytes (OT). Detailed information is summarised in Table 1.

## Ostracod communities
### General observations

Ostracods were found in all localities (and all samples). Living ostracods were found in eight out of ten water bodies in April, ten out of eleven in June, and nine out of ten in September. The relative abundance of bio- and taphocoenoses vary between the localities, and between the months. Compared to the previous months, number of valves and living ostracods were significantly lower in September. An extreme case was documented in a small pond (WS) where number of living ostracods decreased from several hundred (April 794, June 989) to only one specimen in September (Fig. 3). Other localities provide a lower variation in living ostracod numbers (*e.g.*, KS).

### Species list

In total, 27 podocopid ostracod species were identified, twenty of them were also found living.

The following ostracod species were found (based on *Meisch, 2000*):

Superfamily: Darwinuloidea Brady & Norman, 1889
  Family: Darwinulidae Brady & Norman, 1889
      Genus: *Darwinula* Brady & Robertson, 1885
        *Darwinula stevensoni* (Brady & Robertson, 1870)
Superfamily: Cypridoidea Baird, 1845
  Family: Candonidae Kaufmann, 1900
    Subfamily: Candoninae Kaufmann, 1900
      Genus: *Candona* Baird, 1845
        *Candona candida* (O.F. Müller, 1776)
        *Candona neglecta* Sars, 1887
      Genus: *Fabaeformiscandona* Krstić, 1972
        *Fabaeformiscandona fabaeformis* (Fischer, 1851)
        *Fabaeformiscandona holzkampfi* (Hartwig, 1900)
      Genus: *Pseudocandona* Kaufmann, 1900
        *Pseudocandona compressa* (Koch, 1838)
        *Pseudocandona marchica* (Hartwig, 1899)
      Genus: *Candonopsis* Vavra, 1891
        *Candonopsis kingsleii* (Brady & Robertson, 1870)
    Subfamily: Cyclocypridinae Kaufmann, 1900
      Genus: *Physocypria* Vavra, 1897
        *Physocypria kraepelini* G.W. Müller, 1903

Family: Ilyocyprididae Kaufmann, 1900
    Subfamily: Ilyocypridinae Kaufmann, 1900
        Genus: *Ilyocypris* Brady & Norman, 1889
            *Ilyocypris bradyi* Sars, 1890
            *Ilyocypris gibba* (Ramdohr, 1808)
            *Ilyocypris monstrifica* (Norman, 1862)
Family: Notodromadidae Kaufmann, 1900
    Subfamily: Notodromadinae Kaufmann, 1900
        Genus: *Notodromas* Lilljeborg, 1853
            *Notodromas monacha* (O.F. Müller, 1776)
Family: Cyprididae Baird, 1845
    Subfamily: Eucypridinae Bronstein, 1947
        Genus: *Eucypris* Vavra, 1891
            *Eucypris virens* (Jurine, 1820)
            *Eucypris* sp?
        Genus: *Prionocypris* Brady & Norman, 1896
            *Prionocypris zenkeri* (Chyzer & Toth, 1858)
    Subfamily: Herpetocypridinae Kaufmann, 1900
        Genus: *Herpetocypris* Brady & Norman, 1889
            *Herpetocypris chevreuxi* (Sars, 1896)
    Subfamily: Cyprinotinae Bronstein, 1947
        Genus: *Heterocypris* Claus, 1892
            *Heterocypris incongruens* (Ramdohr, 1808)
            *Heterocypris salina* (Brady, 1868)
    Subfamily: Cypridopsinae Kaufmann, 1900
        Genus: *Cypridopsis* Brady, 1867
            *Cypridopsis vidua* (O.F. Müller, 1776)
        Genus: *Plesiocypridopsis* Rome, 1965
            *Plesiocypridopsis newtoni* (Brady & Robertson, 1870)
        Genus: *Sarscypridopsis* McKenzie, 1977
            *Sarscypridopsis aculeata* (Costa, 1847)
        Genus: *Potamocypris* Brady, 1870
            *Potamocypris arcuata* (Sars, 1903)
            *Potamocypris smaragdina* (Vavra, 1891)
Superfamily: Cytheroidea Baird, 1850
    Family: Limnocytheridae Klie, 1938
        Subfamily: Limnocytherinae Klie, 1938
            Genus: *Limnocythere* Brady, 1867
                *Limnocythere inopinata* (Baird, 1843)
    Family: Cytherididae Sars, 1925
        Genus: *Cyprideis* Jones, 1857
            *Cyprideis torosa* (Jones, 1850)

Family: Loxoconchidae Sars, 1925
Genus: *Cytheromorpha* Hirschmann, 1909
*Cytheromorpha fuscata* (Brady, 1869)

## Spatial and seasonal distribution
### *General remarks*

Spatial and seasonal distribution of the ostracod species of the Mansfeld lake area is displayed at Fig. 3. The species list is sorted by maximum salinity tolerance values according to *Frenzel, Keyser & Viehberg (2010)*. This results in five groups, each for the salinity ranges of ≤5, ≤10, ≤15, ≤20, and ≤25, whereby group I refers to the lowest salinity values and group V to the highest. Within a group, species are also sorted by increasing salinity tolerance. Water bodies were also classified in ascending order according to salinity values. Category A includes water bodies outside the former Salziger See area. These water bodies have lower salinity values (0.7 to 4) and are comparatively small. Category B consists of the water bodies within the former Salziger See with higher salinity values (4.4 to 20.7), and are larger than the water bodies in category A (except QW). Category C comprises SS, the largest lake in the study area, and MG, a section of a ditch that encircles the former Salziger See. Both water bodies have low salinity values (0.9 to 1.3). All of the species are freshwater species (except *C. torosa* and *C. fuscata*), most of them are found in group I (lowest salinity tolerance) and just a few species are in groups with higher salinity tolerances (group IV and V).

The water bodies display significant differences in terms of their species composition in bio- and taphocoenoses between the sampling months. The term abundances always refer to relative abundances.

### *April*

Even if species richness is highest in group I, in April almost all species of this group occur only very sporadically and in very low abundances (mostly ≤ 5%). This is particularly clear in biocoenoses and slightly less pronounced in taphocoenoses (Fig. 3).

In group II there is an increase in species richness and abundances in bio- and taphocoenoses. Especially in II A, two species occur in higher abundances (*C. vidua*, *L. inopinata*). In II B, species richness also increases, but abundances are still rather low (mostly ≤5%), especially in taphocoenoses, and also, but much less pronounced, in biocoenoses, due to higher appearances of *H. salina*.

Although group III includes only half as many species as group II, species are equally common and abundant in bio- and taphocoenoses. Here, as well, abundances in bio- and taphocoenoses are significantly higher in III A than in III B, where abundances especially in taphocoenoses are rather low (≤5%). In biocoenoses, the abundances of the species (*C. neglecta, D. stevensoni, P. kraepelini*) are slightly higher than in II B.

Higher abundances in II A and III A are mainly explained by three species (*C. vidua*, *L. inopinata* and *P. kraepelini*), which often and plentifully occur in both, bio- and taphocoenoses. Species belonging to group IV were not found living in April, although *S. aculeata* is the most abundant species in TE and TE2 in taphocoenoses (≤70%).

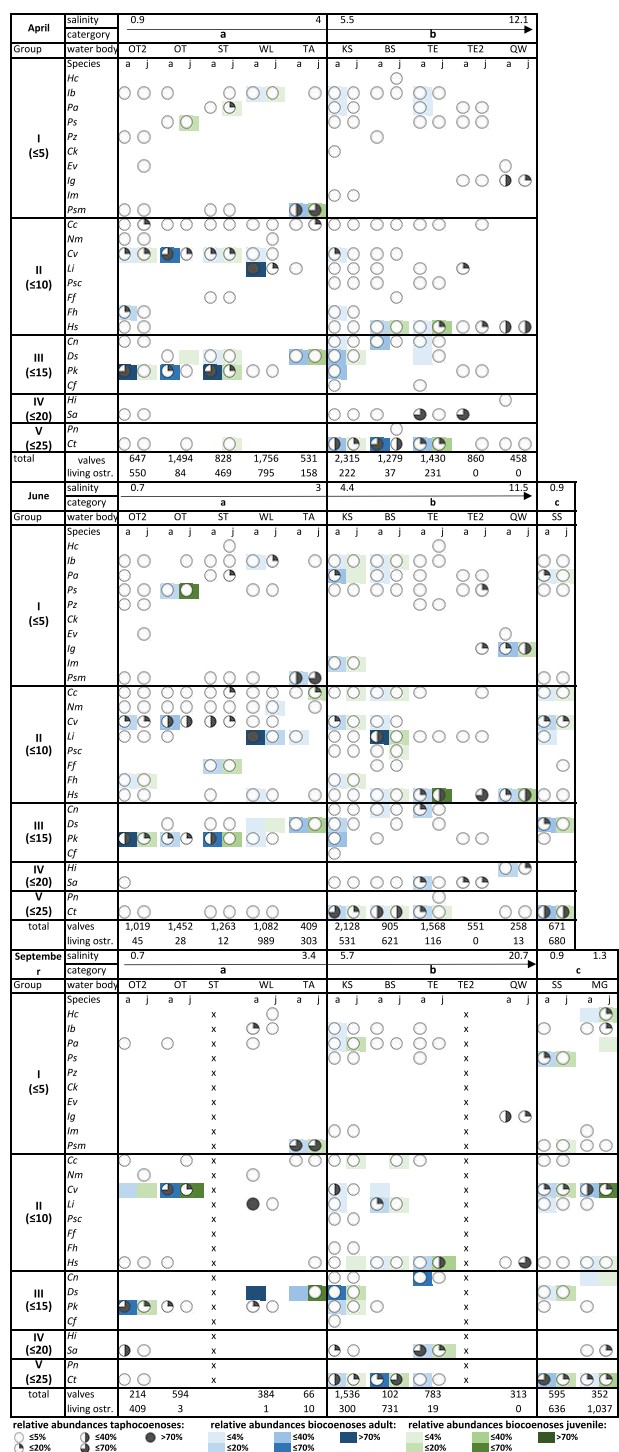

**Figure 3 Relative abundances of taphocoenoses (circles) and biocoenoses (colours) of ostracod species, adult (a) and juveniles (j) in April, June, and September.** Abbreviations of the water bodies: KS, Kernersee; BS, Bindersee; TE, Teufe; TE2, Teufe2; QW, Quelle im Wald; OT, Ottilienteich; OT2, Ottilienteich2; WL, Wannsleben; TA, Tongrube Alte Schule; (continued on next page…)

**Figure 3 (…continued)**
ST, Salzatal; SS, Süßer See; MG, Mittelgraben. Species code: Cc, *Candona candida*; Cn, *Candona neglecta*; Cf, *Cytheromorpha fuscata*. Ck, *Candonopsis kingsleii*; Ct, *Cyprideis torosa*; Cv, *Cypridopsis vidua*; Ds, *Darwinula stevensoni*; E sp?, *Eucypris* sp?; Ev, *Eucypris virens*; Ff, *Fabaeformiscandona fabaeformis*; Fh, *Fabaeformiscandonda holzkampfi*; Hc, *Herpetocypris chevreuxi*; Hi, *Heterocypris incongruens*; Hs, *Heterocypris salina*; Ib, *Ilyocypris bradyi*; Ig, *Ilyocypris gibba*; Im, *Ilyocypris monstrifica*; Li, *Limnocythere inopinata*; Nm, *Notodromas monacha*; Pk, *Physocypria kraepelini*; Pn, *Plesiocypridopsis newtoni*; Pa, *Potamocypris arcuata*; Ps, *Potamocypris smaragdina*; Pz, *Prionocypris zenkeri*; Psc, *Pseudocandona compressa*; Psm, *Pseudocandona marchica*; Sa, *Sarscypridopsis aculeata*.

In V A, occurrences and abundances are very low ($\leq$5% in taphocoenoses and $\leq$4% in biocoenoses), but approximately highest in bio- and taphocoenoses of V B, which is caused by *C. torosa*.

In general, in category a the most common and abundant species in taphocoenoses are the same as in biocoenoses. For instance, *C. vidua* and, *P. kraepelini* are common and abundant, *L. inopinata* occurs locally but with high abundances, and *C. candida* is common but with low abundances and never found alive. In category B, *C. torosa* is the only common and abundant species in bio- and taphocoenoses. Generally, most of the common species are not very abundant (*e.g.*, *C. candida, L. inopinata, H. salina*). Abundant species in taphocoenoses are not abundant in biocoenoses (*H. salina, I. gibba, S. aculeata*) and common and abundant species in biocoenoses are not in taphocoenoses (*e.g.*, *H. salina, C. neglecta, D. stevensoni*).

### June

In June the distribution of the species is much more inconsistent compared to April. In Group I (*i.e.*, low salinity tolerant-species) comparatively few species (*e.g.*, in comparison to group II) can be found again. Nevertheless, there are significantly more species in bio- and taphocoenoses than in April. Species with lowest salinity tolerances (uppermost species of group I) are particularly widespread. This applies to all three categories (A, B, and C). Overall, in this group species are not only more common, there is also an increase in abundances, both in bio- and taphocoenoses. In group II, in all water bodies species are more common, and abundance of some species increases significantly (*e.g.*, *L. inopinata, H. salina*).

In II A, the species distribution mainly relies on the species with comparatively low salinity tolerances (*C. candida, N. monacha, C. vidua*) and the water bodies are nearly similar in their species composition in taphocoenoses. In biocoenoses species richness increase but abundances are rather low (*e.g.*, $\leq$4% for most species). In II B, on the other hand, the differences between the water bodies are larger. Almost all species occur in KS and BS in taphocoenoses and in a lesser degree in biocoenoses, while species are basically not apparent in the other water bodies. An exception represents *H. salina*, which is common and abundant in all water bodies of II B, in bio- and taphocoenoses. Abundances and richness in II C is similar to KS and BS, but a comparison is difficult as this category only includes one water body and was not sampled in April.

In III A are only two of the four species of category a (*D. stevensoni* and *P. kraepelini*). However, these are very common and abundant in bio- and taphocoenoses. In III B the

occurrence is more dispersed and species in bio- and taphocoenoses are not very abundant. In IV A, species richness decrease significantly (*S. aculeata* with ≤4%). In IV B two species (*H. incongruence* and *S. aculeata*) occur with low abundances in bio- and taphocoenoses. In IV C no species was found. Group V is dominated by *C. torosa*, in category A in very low numbers, but in category B and C it is the most common and abundant species in bio- and taphocoenoses.

In general, in all of the three categories abundant species in biocoenoses are also abundant in taphocoenoses (with few exceptions *e.g.*, *C. vidua* in ST and *H. salina* in TE2).

### September

In September, species in group I were found sporadically in the water bodies and only in very low abundances. Except *P. marchica*, which occurs in category a alive and is also very abundant in taphocoenoses.

In I B and C species are slightly more common and more often alive. Also in II A, the species occur only very sporadically, but the abundances in biocoenoses increase slightly (due to *C. vidua*). In taphocoenoses the abundance also increases due to the mass occurrence of *C. vidua* and *L. inopinata*. In II B, species are more common in both bio- and taphocoenoses with a slight increase in abundances in taphocoenoses, even if most of the species (*e.g.*, *C. candida*, *L. inopinata*, *P. compressa*) occur in low abundances (≤5%). In particular, living species (*e.g.*, *C. candida*, *C. vidua*, *L. inopinata*) are common but occur with low abundances (≤4%) in this group and category. In III C, just a few species (*C. candida*, *L. inopinata*, *H. salina*) occur with low abundances, except of *C. vidua* which is the most common and abundant species in bio- and taphocoenoses. In III A are the highest abundances for these water bodies, although absolute numbers of specimens are generally very low. Taphocoenoses are characterized by *P. kraepelini*, which is very common and abundant. In taphocoenoses of III B, all species occur in KS but are nearly absent in the other water bodies and abundances are very low (all ≤5%). In biocoenoses the abundances increase (*C. neglecta* and *D. stevensoni* between 40–70%). In bio- and taphocoenoses of III C abundances are low (≤4% in biocoenoses and ≤5% in taphocoenoses) and species are not very common. In IV A and V A, just two species (*S. aculeata*, adults ≤40% and *C. torosa* with ≤5%) occur in one water body (OT2). In category IV B and C and V B and C, the same two species occur in bio- and taphocoenoses. *C. torosa* is the most abundant species in bio- and taphocoenoses and was found in all water bodies nearly.

Samples from September are characterized by lowest number of species, as *P. zenkeri*, *C. kingsleii*, *E. virens*, *F. fabaeformis*, *H. incongruens* and *P. newtoni* did not occur.

It should also be noted, that in September a relatively large number of species (in comparison to the previous month), not occurring in taphocoenoses are found in biocoenoses (e.g., *C. vidua*, *D. stevensoni*, *C. neglecta*). However, occurrences in the water bodies are similar to previous samplings.

In general, in all waterbodies most of the species that are abundant in taphocoenoses are also abundant in biocoenoses. But there are more exceptions in September than in the previous months. In category a *C. vidua* for example is abundant species in OT2 in biocoenoses, but is not present in taphocoenoses, *L. inopinata* is the most abundant

species in taphocoenoses in WL, but living specimens did not occur, instead *D. stevensoni* is dominant (but just with one living specimen) in biocoenoses, but is not present in taphocoenoses.

In category B most of the species that are abundant in biocoenoses are less abundant in taphocoenoses (≤4%), except *C. torosa* which is abundant in both.

In category C abundances of bio- and taphocoenoses correspond most closely.

### In summary

Comparing the ostracod species of categories A and B (*i.e.,* low *vs.* higher salinities), two patterns emerge. First, two species (*P. marchica* and *N. monacha*) occur exclusively in bio- and taphocoenoses of category A (low salinity), and two species (*P. compressa, I. gibba*) appear only in bio- and taphocoenoses of category B (higher salinity). All of these four species are recorded with low specimen numbers only. Another point relates to the abundances of species. Some species occur in both water body categories, but are common and abundant in one category only, while they occur sporadically and with low abundances (and mostly only in taphocoenoses) in the other category. Although there are fluctuations in the abundances of the water bodies between the months (especially in the absolute values), it is noticeable that two species of the biocoenoses (*C. vidua* and *P. kraepelini*) are common and mostly dominant in the water bodies with lower salinities (category A) in all months. The same applies for the water bodies with higher salinity (category B), the species here are *H. salina* and *C. torosa*. The same pattern can be observed in taphocoenoses.

Despite the lower salinity values, the ostracod assemblages of category C differ from category A. Peculiarly, the ostracod assemblages of category C represent a combination of categories A and B, due to the occurrence of the most abundant species *C. vidua* and *C. torosa*.

There is another aspect, differentiating the water bodies from each other. There is a lower species richness in smaller water bodies, dominated by one species while other species provide only low abundances (*e.g.,* in OT2, *P. kraepelini* exaggerates with 85% the other two species *F. holzkampfi*; 9% and *C. vidua*; 6% by far). This becomes particularly obvious in biocoenoses and is very stable during all months.

Contrary, in larger water bodies, species richness is significantly higher, while no species is clearly dominant and varies during the months. The ambiguous distribution of species can also be seen in the cluster analyses of water bodies and species (Fig. 4). The cluster analysis of the biocoenoses reveals that the similarity of the water bodies depends on the seasonal distribution of the species. Each sampling (*i.e.,* month) provides slightly different species composition contributing to distinct species assemblages. Thus, taken all together there are neither distinct types of water bodies (and related environmental conditions) nor specific living ostracod species assemblages (Fig. 4C, 4D).

The picture is somewhat clearer in taphocoenoses of the seven water body groups (Fig. 4A). More or less the three samplings of a water body form a group, mostly together with spatially close (or connected) water bodies (*e.g.,* KS, BS, MG, SS are one group). Only WL and TA form separate groups, but are also more spatially distant from the other water bodies, and thus are more isolated. Cluster analysis of the species revealed only two groups

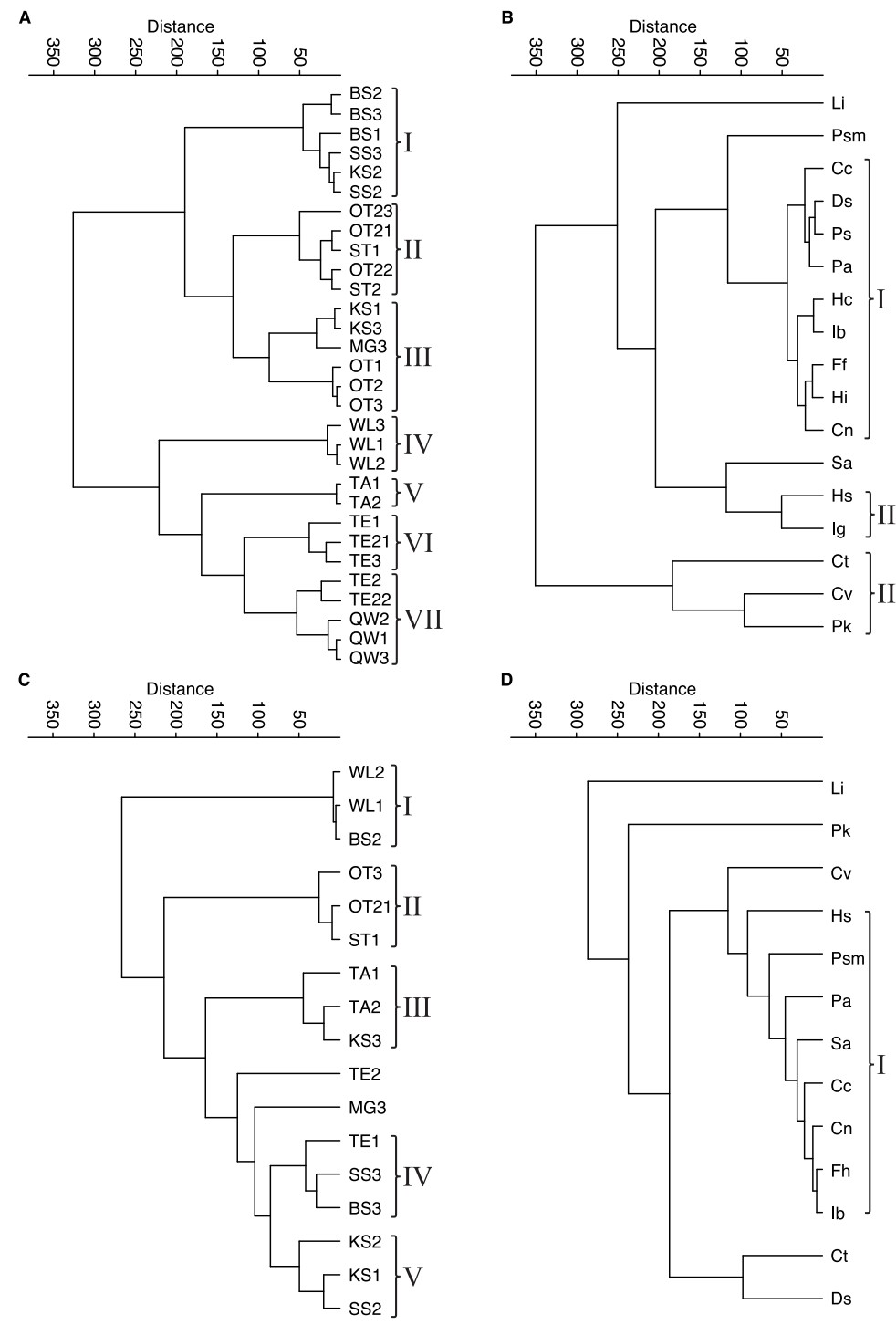

**Figure 4  Cluster analyses of water bodies and species for taphocoenoses and biocoenoses.** (A) taphocoenoses water bodies, (B) taphocoenoses species, (C) biocoenoses water bodies, (D) biocoenoses species. (continued on next page...)

**Figure 4 (…continued)**
Abbreviations of the water bodies: KS, Kernersee; BS, Bindersee; TE, Teufe; TE2, Teufe2; QW, Quelle im Wald; OT, Ottilienteich; OT2, Ottilienteich2; WL, Wannsleben; TA, Tongrube Alte Schule; ST, Salzatal; SS, Süßer See; MG, Mittelgraben. Species code: Cc, *Candona candida*; Cn, *Candona neglecta*; Ct, *Cyprideis torosa*; Cv, *Cypridopsis vidua*; Ds, *Darwinula stevensoni*; Ff, *Fabaeformiscandona fabaeformis*; Hc, *Herpetocypris chevreuxi*; Hi, *Heterocypris incongruens*; Hs, *Heterocypris salina*; Ib, *Ilyocypris bradyi*; Ig, *Ilyocypris gibba*; Li, *Limnocythere inopinata*; Pk, *Physocypria kraepelini*; Pa, *Potamocypris arcuata*; Ps, *Potamocypris smaragdina*; Psm, *Pseudocandona marchica*; Sa, *Sarscypridopsis aculeata*. Roman numbers represent the different species- and water body groups.

that maybe can be associated with an environmental parameter (Fig. 4B). These are *P. kraepelini* and *C. vidua* occurring in lower saline water bodies as well as *I. gibba* and *H. salina* occurring in water bodies with higher salinities. However, again, most species (nine) are in one group showing no species-specific preferences and four species (*L. inopinata, P. marchica, S. aculeata* and *C. torosa*) are separate.

## Seasonal population structure

For the seasonal population structure combined data from all species (adult: juvenile of living: dead ratios of assemblages) collected at each sampling site are considered (Fig. 5). There is no generalized pattern in distribution of adults and juveniles in taphocoenoses and biocoenoses in the water bodies. Ratios of relative abundances for living: dead and adult: juveniles were different for the water bodies and between the sampling months. Some localities show relatively constant ratios through the sampling period (*e.g.*, KS, TE, SS), while others vary greatly (*e.g.*, BS, TE2). In most water bodies the number of valves exceeds the number of living ostracods found (KS, TE, OT). Significantly, more living than valves were found in some waterbodies (and months) (BS in September, OT2 in April and September, MG). In taphocoenoses, some water bodies provided significantly more adults than juveniles (*e.g.*, KS, WL) while juveniles exaggerate adults in other water bodies (*e.g.*, QW, TA). In SS approx. equal numbers of living and valves were found. MG is the only water body, in which significantly more juveniles than adults were found.

## Species diversity

For diversity and richness data, taphocoenoses for all three months were considered together. Species diversity (*i.e.,* Fisher's $\alpha$ diversity) and richness show the same trend (Fig. 6), whereby the richness shows partially lower values. Richness and Fisher's $\alpha$ diversity also do not show a clear trend or pattern in relation to salinity. Up to 4, species richness and diversity decrease with increasing salinity, following the pattern from *Frenzel (2009)*. From 4, however, richness increase, and has its maximum at 6. Diversity of species decreases again at values higher than 6.

## Inferences on species life cycle

Although only three samplings were made, covering only half a year, living ostracods show clear differences in their abundance patterns (Fig. 7). The time at which maximum and minimum abundance of the species is attained, differs for most species and for the water bodies. For instance, *C. torosa* shows a peak in April (KS, TE) and September (BS), minimum abundance is in June in two water bodies (KS, BS) while the abundance is

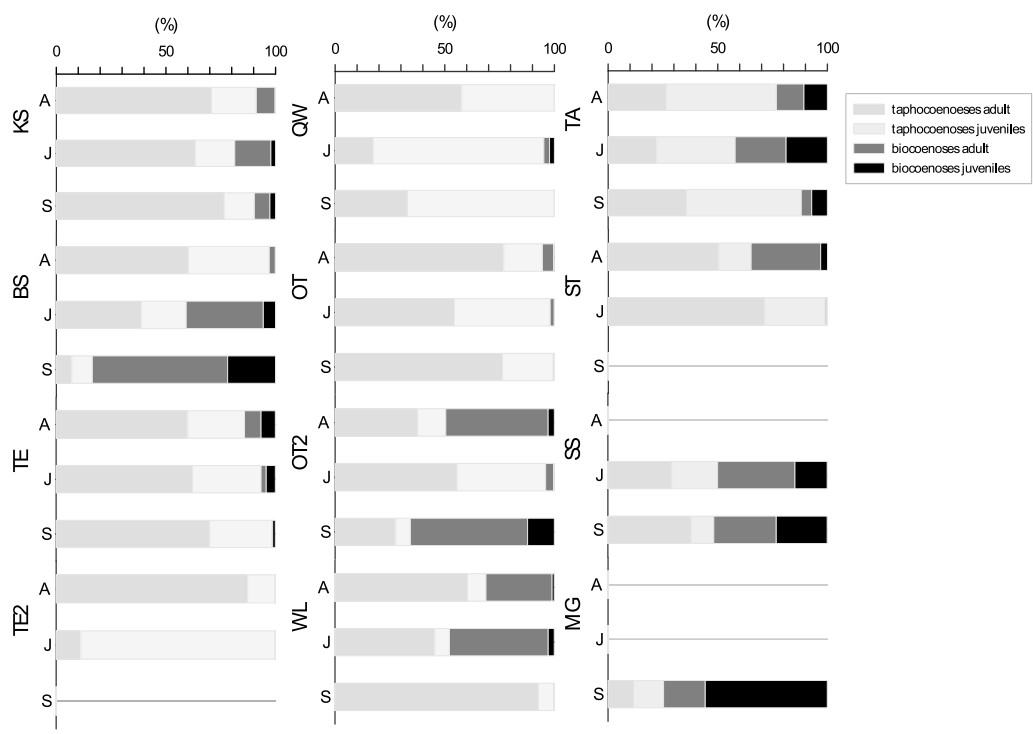

**Figure 5** **Ratios of adult and juvenile ostracod assemblages differentiated for bio-and taphocoenoses.** Bars represent combined data from all species collected in each water bodies in April (A), June (J) and September (S). Abbreviations of the water bodies: KS, Kernersee; BS, Bindersee; TE, Teufe; TE2, Teufe2; QW, Quelle im Wald; OT, Ottilienteich; OT2, Ottilienteich2; WL, Wannsleben; TA, Tongrube Alte Schule; ST, Salzatal; SS, Süßer See; MG, Mittelgraben.

nearly invariable low in June and September in another site (TE). While few juveniles were generally found in KS, they have maximum abundance in April (TE) and September (BS). An extreme case is displayed by *L. inopinata*, which is highly abundant in WL (>90% of the entire species assemblage) in April and June, but disappear completely in September. In BS *L. inopinata* did not occur in April, but was very abundant in June (>80%) and occur in lower abundances in September (10–20%). While *C. vidua* show the same abundance pattern in OT and OT2 (peak in September, minimum in June), relative abundances of these two water bodies vary extremely (OT 35–100%, OT2 4–25%). *Physiocypris kraepelini* has a peak in April (OT) and June (KS, OT2) and a minimum in September in all three water bodies (in OT no individuals were found in September). For the species *H. salina*, generally few adult individuals were found (15%), both in BS, and in TE. Peaks are in April (BS) and June (TE). *Darwinula stevensoni* shows a significant minimum in June (in KS), otherwise the abundances of this species are high. Peaks are in September (KS and TA), but for TA due to the high numbers of juveniles. The maximum of adults for TA is in June. *Heterocypris salina* (TE) and *D. stevensoni* (September in TA) are the only species of which more juveniles than adults were found. Each species displays a different abundance distribution over the months in the water bodies. To obtain a general trend, the total species abundances of all waters were considered together (Fig. S1). For example,

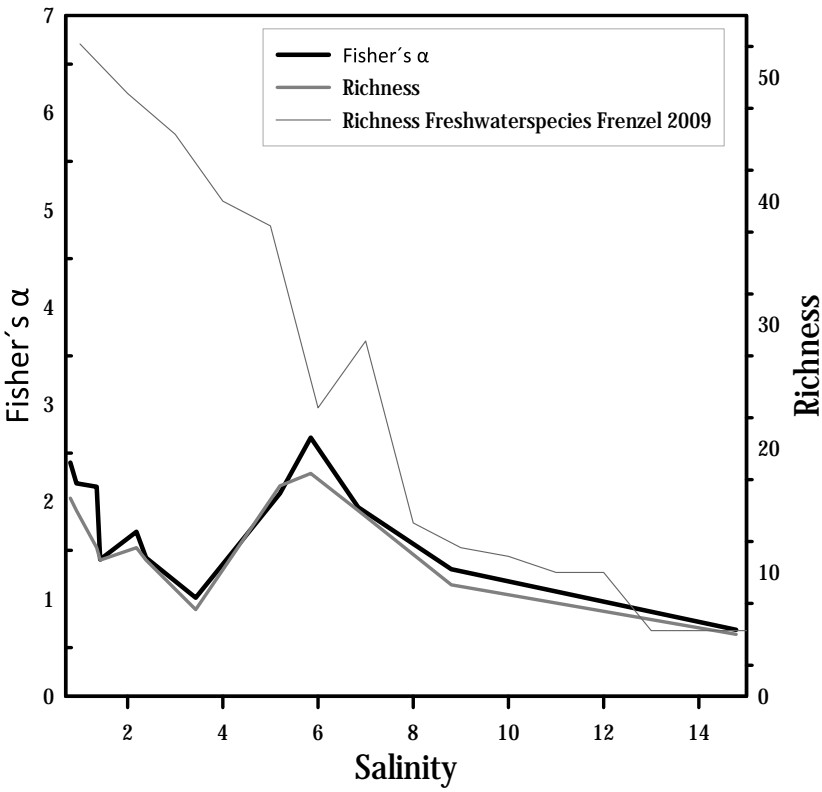

**Figure 6** Fisher's *α* diversity and richness of cumulated taphocoenoses of all sampled month of the water bodies at Mansfeld area in comparison with richness of freshwater species in waterbodies in the catchment of the Baltic Sea *Frenzel (2009)*.

the total abundances show, that three species (*D. stevensoni, H. salina, L. inopinata*) are most abundant in June, while three (*C. torosa, C. vidua, P. kraepelini*) species have their minimum in June. Moreover, conclusions about the preservation potential of the valves of these species are possible by additional examination of the taphocoenoses of each species. In *C. torosa*, for example, the number of valves exaggerate living ones, which is also displayed by *H. salina, C. vidua*, and to a lesser degree in *P. kraepelini*. In *L. inopinata* and *D. stevensoni*, on the other hand, more living specimens than valves were found.

## Species distribution and environmental conditions

Living ostracods occur in the Mansfeld lakes at salinity range between 0.7 and 11.5 (Fig. 8). Most species are associated with values between 1 to 6. Only three species (*H. incongruens, H. salina* and *I. gibba*) occur at values up to 11.5. *H. salina* shows the highest tolerance to salinity (1.4–11.5).

Living ostracods were found in a temperature range between 9.4 up to 26.5 °C. On average, most species were found between temperatures from 10 to 24 °C. Five species (*C. candida, F. fabaeformis, I. monstrifica, L. inopinata, N. monacha* and *S. aculeata*) occur in warmer water bodies (>18 °C). The dissolved oxygen values, at which living ostracods were found, range between 4.3 mg/l and 19.4 mg/l. Majority of species occur between 5

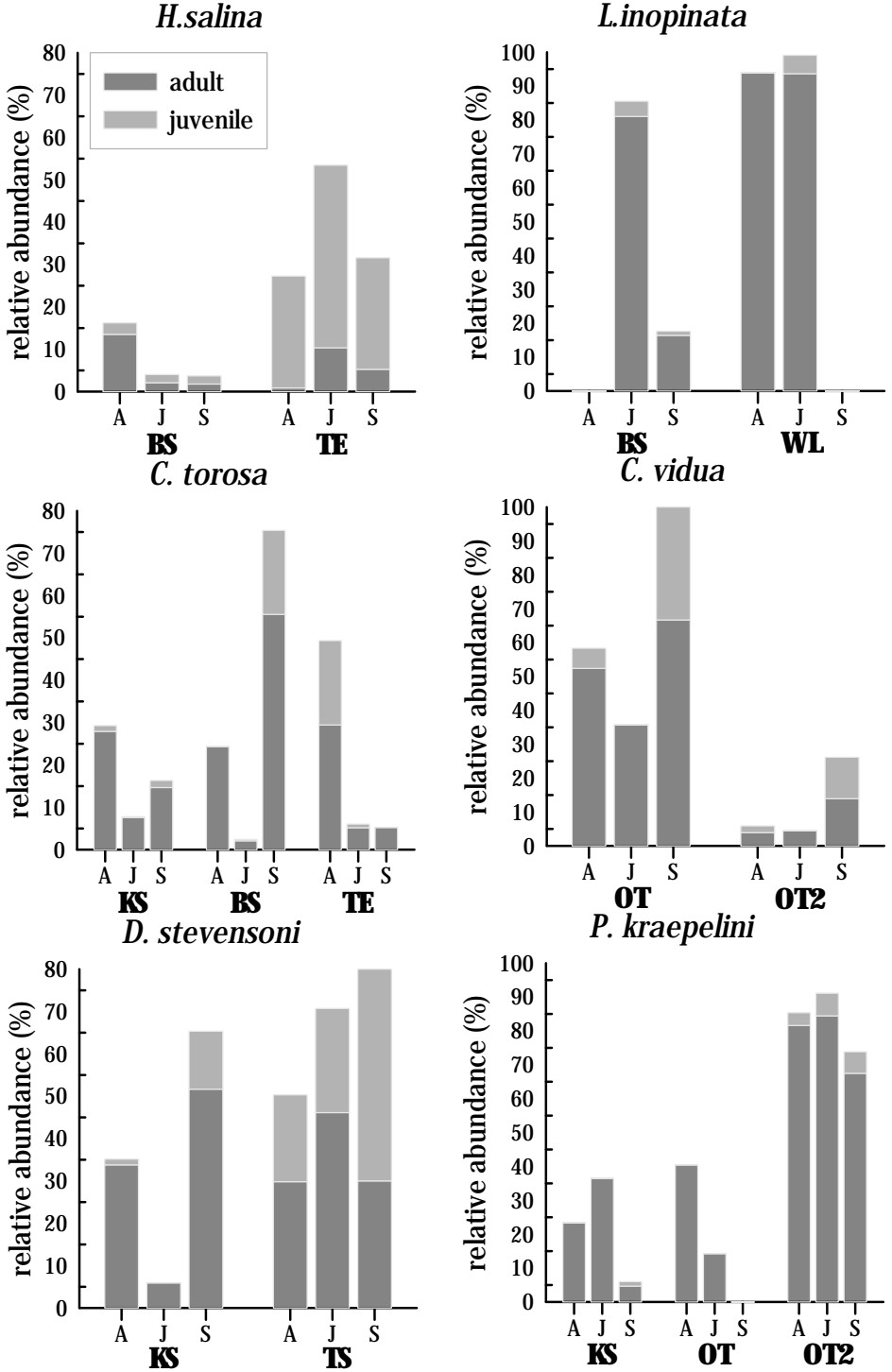

**Figure 7  Seasonal distribution of selected most common and abundant living species in selected waterbodies differentiated for adult and juveniles in April (A), June (J) and September (S).** Abbreviations of the water bodies: KS, Kernersee; BS, Bindersee; TE, Teufe; TE2, Teufe2; QW, Quelle im Wald; OT, Ottilienteich; OT2, Ottilienteich2; WL, Wannsleben; TA, Tongrube Alte Schule; ST, Salzatal; SS, Süßer See; MG, Mittelgraben.

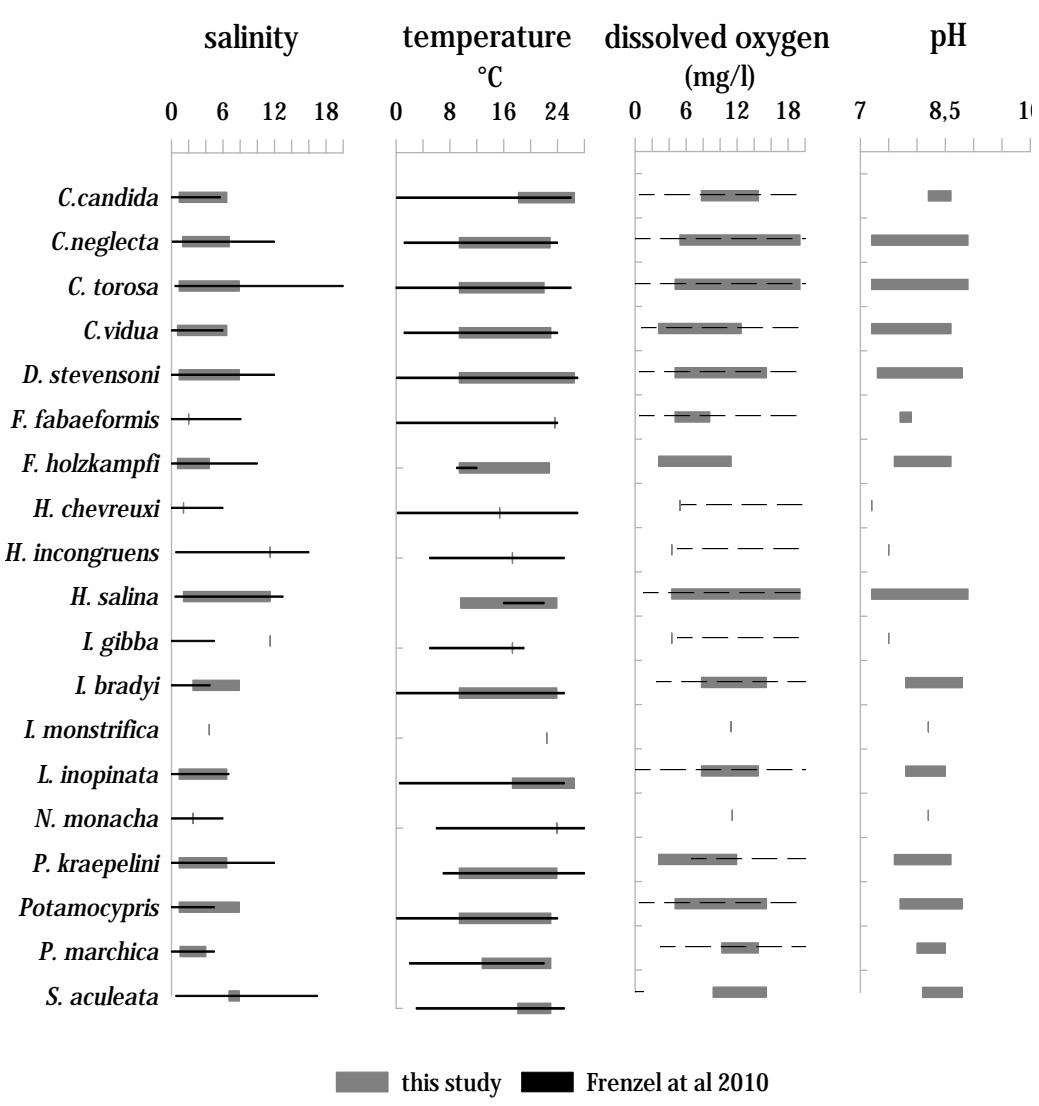

Figure 8 **Distribution of ostracod species in relation to variation ranges of salinity, temperature, dissolved oxygen, and pH inferred from all samplings and localities compared to the study of** *Frenzel, Keyser & Viehberg (2010)*. Solid lines represent a known range, dashed lines represent only maximum values are known.

and 16 mg/l. Although the range of most species is high, there are more species associated with lower dissolved oxygen values. Only three species (*C. neglecta, C. torosa* and *H. salina*) were found living at values over 19 mg/l.

The pH range under which living ostracods were found is between 7.2 and 8.9. Most species are present between 7.5 and 8.6. Three species (*C. neglecta, C. torosa* and *H. salina*) occur in the entire pH range.

The canonical correspondence analysis plot (Fig. 9) displays the relationships between physico-chemical parameters, ostracod species and localities. The first axis explains with 60% most of the variation and can be correlated with the parameters pH, temperature and

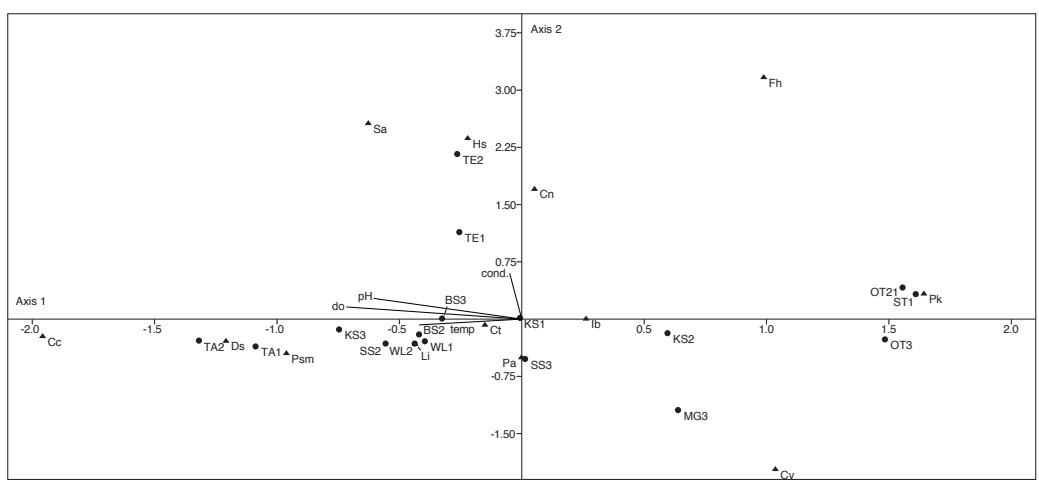

**Figure 9 First two axes of canonical correspondence analysis (CCA) ordination plot of ostracods (▲), environmental variables (cond, conductivity; do, dissolved oxygen; tem, temperature and pH) and water bodies (●).** Abbreviations of the water bodies: KS, Kernersee; BS, Bindersee; TE, Teufe; TE2, Teufe2; QW, Quelle im Wald; OT, Ottilienteich; OT2, Ottilienteich2; WL, Wannsleben; TA, Tongrube Alte Schule; ST, Salzatal; SS, Süßer See; MG, Mittelgraben. Cc, *Candona candida*; Cn, *Candona neglecta*; Ct, *Cyprideis torosa*; Cv, *Cypridopsis vidua*; Ds, *Darwinula stevensoni*; Fh, *Fabaeformiscandonda holzkampfi*; Hs, *Heterocypris salina*; Ib, *Ilyocypris bradyi*; Li, *Limnocythere inopinata*; Pk, *Physocypria kraepelini*; Pa, *Potamocypris arcuata*; Psm, *Pseudocandona marchica*; Sa, *Sarscypridopsis*. Sample months are represented by numbers (1 = April, 2 = June and 3 = September) following the water body abbreviation.

dissolved oxygen. The second axis explains 22.5% of the variation and can be correlated with conductivity. However, only a few species show a significant correlation with the measured parameters (Table S2).

*D. stevensoni* correlates with pH ($r = 0.56$, $p = 0.02$) and dissolved oxygen ($r = 0.57$, $p = 0.02$), *L. inopinata* with temperature ($r = 0.53$, $p = 0.03$) and *P. marchica* ($r = 0.54$, $p = 0.02$) with dissolved oxygen. Two species correlate with salinity, *H. salina* ($r = 0.51$, $p = 0.03$) and *S. aculeata* ($r = 0.57$, $p = 0.02$). Nevertheless, no (habitat specific) groups can be distinguished in the CCA.

## DISCUSSION

### General aspects on ostracod species distribution

All ostracod species found are reported as cosmopolitan, euryoecious freshwater species, and are typical for central Europe (*Fuhrmann, 2012*; *Meisch, 2000*), and in particular for the region Mansfeld (*Pint et al., 2015*; *Wennrich, 2005*). Due to the natural salinity of the lakes, the occurrence of the two brackish water species *C. fuscata* (only valves) and *C. torosa* are not uncommon and already previously documented (*Pint et al., 2012*; *Wennrich, 2005*). So far, no living *C. torosa* from (saline) inland occurrences in Germany are known, although valves of *C. torosa* have been found in higher saline water bodies in central Germany (*Pint et al., 2012*; *Scharf, Herzog & Pint, 2017*). The disappearance of *C. torosa* in Mansfeld area was explained by the draining of the Salziger See in 1892 (*Fuhrmann, 2012*; *Scharf, Herzog & Pint, 2017*) and also *Pint et al. (2012)*; *Pint et al. (2015)* were not able to find

living *C. torosa* in the region. Therefore, the high numbers of living individuals of *C. torosa* found, especially in lower saline water bodies, are remarkable.

With 27 species found in the Mansfeld area, with a maximum of 17 species in one site (and maximum of nine living), the number of species is relatively high with regard to the size of the study area and the small size of the water bodies. Size of the region (and number of water bodies) is decisive for the number of species found (*Altınsaçlı, 2001*; *Kulkoyluoglu, Sari & Akdemir, 2012*; *Rossetti, Bartoli & Martens, 2004*). Species richness and diversity of a water body depends on size of the water body (*Marchegiano et al., 2017*; *Rossetti et al., 2006*; *Valls et al., 2016*), connectivity of water bodies, type of habitat and related environmental conditions (*Altınsaçlı, 2001*; *Kulkoyluoglu et al., 2018*; *Kulkoyluoglu & Vinyard, 2000*) and an interplay of different factors, such as substrate type, vegetation, food availability, season and water depth (*Smith & Delorme, 2010*). The more ecological niches a water body has, the more species can be found (*Iglikowska & Namiotko, 2012*). A comparison with studies from marginal marine area also demonstrates that diversity and richness are dependent on the size of the study area (*Frenzel, 2009*; *Remane, 1934*). The fluctuations between the water bodies of diversity and richness are significantly lower, and the trend observed in the marginal marine area of a decrease in richness with increasing salinity is not observed in the Mansfeld area. This shows that the relationship between richness and salinity is scale-dependent. High species richness usually indicates undisturbed habitats and stable environmental conditions (*Carbonel et al., 1988*; *Valls et al., 2016*).

For an evaluation, however, species composition must also be considered. As previously mentioned, most of the species occurring in the Mansfeld lake area are considered as widespread generalists with regard to their ecological preferences (in particular the common and abundant species). It is stated that cosmopolitan species often occur in disturbed habitats with low water conditions (*Ghaouaci et al., 2017*; *Külköylüoğlu, 2013*). Our data confirm this assumption, especially at the Süßer See, and at the residual lakes of the former Salziger See. Due to mining activities in the region during the last 800 years, soil and waters are exposed to high levels of pollutants from geological sources, but also anthropogenic pollutants from mine tailings and smelting products of copper shale mining (*Becker et al., 2001*; *Frühauf, 1999*). In particular, Süßer See is considered as a sink for heavy metals (*Becker et al., 2001*). In addition, there are also considerable nitrogen and phosphate inputs from intensively agricultural use of the region (*Lewandowski, Schauser & Hupfer, 2003*; *Schmidt, Frühauf & Dammann, 2010*).

## Spatial distribution of ostracod species

Although the occurring ostracod species are reported to have wide and rather unspecific ecological preferences, each water body is characterised by specific ostracod species assemblages. Despite some water bodies are more similar (*e.g.*, water bodies of former Salziger See) in their assemblages than others (*e.g.*, water bodies without former Salziger See), each water body shows specific species composition (and species richness) and abundances during the sampling period. These differences can be explained by different biotic and abiotic conditions (*Smith & Delorme, 2010*). Cluster analysis of the

taphocoenoses revealed that similarities could originate from the hydrological connection of some water bodies (SS, KS, BS and MG).

The differentiation between living ostracods and valves displayed that species relative abundance and richness is much higher in valves, which might related to the different temporal scales integrated by these two associations (*Akita et al., 2016*; *Valls et al., 2016*). Biocoenoses represent short-term population dynamics and associated environmental conditions at the time the sample was taken and are therefore strongly influenced by the seasonality or life-cycle, respectively, of the species (*Winegardner et al., 2015*). Taphocoenoses, on the other hand, integrate, through accumulation and time averaging, several generations over seasons and years and species which are rare or absent in biocoenoses (*e.g.*, *P. zenkeri, E. virens*) and accumulate, thus, a species assemblages corresponding to a larger range of environmental fluctuations (*Levi et al., 2014*; *Poquet et al., 2007*). Taphocoenoses not only contain ostracods through time, but also intra- and inter-habitat migrations (by water fowls or wave motion) are captured (spatial factor) (*Mezquita et al., 2005*; *Winegardner et al., 2015*). The Mansfeld lake area is not only a bird sanctuary with numerous water fowl species, it is also a staging and wintering site, water fowls and bird's may influence the intra- and inter-habitat dispersal of the species (*Al Hussein et al., 2000*). Also an air-born colonization of species by migratory birds, as it is also assumed for the foraminifer genus *Ammonia* and the ostracod species *C. fuscata*, is possible (*Dieffenbacher-Krall, 2013*; *Wennrich, Meng & Schmiedl, 2007*). This can lead to a distortion of the results of species distribution and related environmental inferences (*Dieffenbacher-Krall, 2013*).

The strong differences of species abundances highlight that the location of sampling within a water body is crucial. Sampling positions only a few meters away from each other can provide significantly different live and dead species assemblages which might be caused by differences in microhabitat conditions (*Decrouy, 2009*) and/or very local transport mechanisms. This could be a reason why no living *C. torosa* has been found in the past few decades. The occurrence of single ostracod species or populations in a water body can be restricted to very local areas (*Marchegiano et al., 2017*; *Smith & Delorme, 2010*) and is not mandatorily related to changes in water chemistry (*i.e.,* temperature or salinity) as it is often assumed as in the case of *C. torosa* (*Pint et al., 2012*). In order to obtain a comprehensive picture of the ostracod fauna, multiple sampling should therefore ideally be carried out. If only one sampling can take place at a site, this site should be carefully selected in order to better classify the results.

### Seasonal distribution of ostracod species

The differences in abundances and occurrences of species in the samples show, that not only sampling site but also the time (*i.e.,* specific month) of sampling have an important influence on the distribution and abundance of living species due to their specific life-cycles (*Altınsaçlı, Perçin-Paçal & Altınsaçlı, 2015*; *Decrouy, 2009*). Thereby, the species show an individual pattern of occurrence in each water body. *Heip's (1976)* extensive investigations on the life cycle of *C. torosa* have shown that this species produces one generation per year, with a minimum in April, followed by increase of specimens, with a maximum peak

between July and October. Even if in the Mansfeld area the maximum in the water bodies varies (KS and TE in April, BS in September), all water bodies show a minimum in June. Also, considering the summarized relative and absolute abundances for all considered water bodies, a significant minimum is found in June (Fig. S1). Although the peak in September corresponds with Heip's observations, the population should increase from April and not decrease progressively in June. Also the high adult abundances in BS in September are remarkable, considering the low abundances of juveniles in June.

*Cypridopsis vidua, D. stevensoni, H. salina* as well as *L. inopinata* are reported to occur throughout the year, but are most abundant in the summer months (from May to October-November) (*Meisch, 2000*). In the Mansfeld area, all species can be found already in April, and partly in considerable numbers (especially *L. inopinata*). *L. inopinata* develops different abundances and overwintering strategies depending on whether they live in freshwater or saline water bodies and these populations never co-occur. Freshwater populations appear in April/May and disappear in October/November, while saline population overwinter (*Geiger, Otero & Rossi, 1998*). Variations in the temporal occurrence of *L. inopinata* in different water bodies (BS and WL) were also observed in the Mansfeld area. Whether these are caused by the different salinities (BS: 6.4, WL: 2.1) of the water bodies is uncertain, since no *L. inopinata* was found in BS in April, and also the disappearance of *L. inopinata* in WL in September cannot be explained. It should also be mentioned that in WL one male carapace was found, what probably indicates a rare male population (one male among about 1,000 female) (*Meisch, 2000*).

The fact that *H. salina* produces two to three generations per year (over summer month) and has relatively short life cycles (about 45 days) could explain the high abundance of juveniles in all months (*Meisch, 2000*). Although *H. salina* has such short life cycles, comparatively few valves are found in the sediment (and predominantly juveniles). Taphocoenoses raw data show that *H. salina* was found relatively often as carapace in all water bodies in the area. For juveniles this suggests high mortality. Especially in TE the number of carapace is high, predominantly in June. This may be an indication of non-optimal conditions, and it suggests that valves and carapaces were less affected by transport in TE, compared to BS. The lifespan of *D. stevensoni* can range from <1 to 4 years, depending on the temperature (*Van Doninck et al., 2003*). Studies in a water body in Belgium resulted in a life span of <1 year, with reproduction starting in April (completed in September), with a maximum in June/July and lower densities in winter (*Van Doninck et al., 2003*). The minimum in June of *D. stevensoni* in KS indicates a different population dynamic in this water body. In TA, the maximum in June fits the results from Belgium, but the high numbers of adults and especially juveniles in September are remarkable. Temperature of the water bodies (KS: 10–23 °C, TA: 13–26 °C, Belgium pond: March to August: 10–24 °C) is nearly invariable, and therefore not sufficient to explain the heterogeneous population dynamics, in this case. Whether the life cycle of *D. stevensoni* is postponed, or lasts longer than that observed in Belgium, cannot be clarified using the data gathered.

Populations of *P. kraepelini* are reported to remain constant in size throughout the year (*Meisch, 2000*). In the Mansfeld area, however, there are fluctuations, with *e.g.*, the

drastic decrease of specimens in KS and OT, but also a general decrease of the population in June. The life cycle of some species could have been adapted to the climatic conditions (warm-toned mesoclimate) in the study area and, therefore, begins somewhat earlier or is postponed (*D. stevensoni*, *C. torosa*). In the case of *C. torosa* it could also be possible that there is more than one generation (possibly two) per year, as it is known from Mediterranean populations (*Mezquita, Olmos & Oltra, 2000*).

Although only three samplings were carried out during the summer months at intervals of two to three months, some deviations in the life cycle of the species, with regard to the temporal occurrence of the species and their minimum and maximum abundance, are evident. However, conclusions on possible causes of these deviations, detailed studies covering a longer time period are required, which should be carried out at shorter intervals in order to record species life cycle with shorter life spans (*e.g.*, *H. salina* 45 days *Meisch, 2000*).

## Inferences on taphonomical processes

Inferences can be drawn about the productivity and preservation potential of the particular sampling site considering the ratio of living to dead ostracods (*Kidwell, 2013*). Water bodies, valves outnumber livings provide higher productivity and preservation potential (*e.g.*, KS, TE, OT). If the valves outnumber livings, or is nearly the same, this may have several causes. First, sampling time has an impact on species productivity (depending on the life-cycle) and hydrological conditions (*e.g.*, extreme rainfall washes away valves) may also matter (*Avnaim-Katav et al., 2021*; *Jorissen & Wittling, 1999*). This is evident in the example of BS, where not only the number of livings (April: 1279 valves and 37 living; September: 102 valves and 731 living) increase extremely, but also the number of valves significantly decrease. Second, the habitat type, with regard to different water energy levels, is important (*Frenzel & Boomer, 2005*). While almost all the water bodies are lakes and ponds sampled in the shallow littoral, MG is a ditch sampled on its slope. Since here the living outnumbers the valves, it can be assumed that due to the slope and flow of the water most of the valves are removed from the sample site. Also, the substrate type and vegetation may also be important, as it also significantly affects the sample size (*Danielopol et al., 2002*). In most water bodies the substrate type is mud or sand. In SS, also sand was the substrate type, but the littoral bottom was covered with coarse gravel and large stones, which made sampling much more difficult. Thus, much smaller amounts of sediment were sampled, which probably influenced the ratio of living to valves in favour of the livings. In general, the sample composition was very different due to substrate type (mud, sand, gravel) and vegetation (reed-belt, die-back reed-belt and other plant remains). Additionally, the valves can have different preservation potential due to differences in hinge type, valve thickness and/or shape (*Alin & Cohen, 2004*; *Avnaim-Katav et al., 2021*). Some living species were not found in the taphocoenoses of the respective water bodies (*e.g.*, *C. vidua* and *D. stevensoni*). Especially valves of *D. stevensoni* are very thin and fragile and may have been relocated or destroyed (*Meisch, 2000*). *Cyprideis torosa* on the other hand, has relatively large, thick valves, which are therefore not transported so quickly and are easily preserved (*De Deckker & Lord, 2017*). Even if the number of living *C. torosa* is partly

small in some water bodies, the valves can be found *en masse* in the sediment. Thus, the preservability of the valves also plays a role in which species (and in which abundances) are found.

However, not only the ratio of valves to living revealed informations about the conditions, also the ratio of adults to juveniles of the taphocoenoses, can be used to conclude about taphonomic processes (*Boomer, Horne & Slipper, 2003*). In most water bodies the adults are by a multiple higher than juveniles by (KS, BS, TE, OT, OT2, WL, ST, SS). Only two water bodies provided similar ratios (MG) or number of slightly higher juveniles than that of adults (TA). And only in two water bodies at one sampling time (TE in June and QW in June) juveniles outnumbers adults significantly. Again, the differences between the months of a water body probably have seasonal causes (*e.g.*, lake level fluctuations, due to precipitation or evaporation). A higher number of adults in the taphocoenoses, and conversely a lower number of juveniles (especially the first instars), suggests post-mortem processes such as transport, relocation and destruction of the valves (*Boomer, Horne & Slipper, 2003*). As the littoral of a waterbody is most affected by currents, wave motion, terrestrial run-off and seasonal water-level fluctuation (observed in WL and SS for example) (*Gasith & Gafny, 1990*; *Peters & Lodge, 2009*), taphocoenoses typically are composed mainly by adults and late juveniles, while early instars are resuspend to deeper waters (*Zhai et al., 2015*). Due to the location in the central German dry region, the water bodies are highly affected by long dry periods (with low water levels) and short extreme precipitation events (and high run-offs with high nutrient inputs) (*Schmidt, Frühauf & Dammann, 2010*). Assuming strong contrasts between dry periods and precipitation (events) which may associated with reworking of littoral sediments the high number of adults compared to juveniles is not exceptional. It would also explain why almost no valves were found in sieve residues <125 µm. Additionally, some species are represented by valves only in very low abundances and single locations (*e.g.*, *P. zenkeri, E. virens*). These factors (*e.g.*, only single valves from a species, higher adult: juvenile ratio) indicate, not only that the assemblages are strongly influenced by taphonomic processes, they also indicate a disturbed habitat and could be stressors for the species. (*Padisak, 1993*).

Including the 125 µm fraction and differentiating juvenile stages could have been provide information, which components of each assemblage are most affected by transport and size sorting (*Boomer, Horne & Slipper, 2003*).

## Ecological inferences

According to *Fuhrmann (2012)* and *Meisch (2000)*, most of the species found prefer warm stagnant or cool stagnant water bodies. However, this classification is not sufficient to explain the heterogeneous spatial distribution of the species in this study. The area is spatially very limited, deviations of abiotic parameters of the water bodies are relatively small and almost all species are assumed to have large tolerance ranges for (measurable) physico-chemical parameters. Thus, all species occurrences reflect the known range of physico-chemical parameters (*Frenzel, Keyser & Viehberg, 2010*; *Ruiz et al., 2013*). Only *I. bradyi* was found living in higher saline waterbodies (up to 7.9) than the known range

from literature (4.5). The higher range of *Potamocypris* is probably due to the mixing of two species resulting from the difficulty in distinguishing them from each other.

We need to inquire why not all species were found in all water bodies. This may have several reasons. Sampling of the water bodies may not cover the entire ostracod fauna of a water body (*Poquet & Mesquita-Joanes, 2011*), as probably indicated by higher numbers of species in the taphocoenoses and the deviations of species composition and abundances in bio- and taphocoenoses.

Water bodies could provide, *e.g.*, due to different hydro(geo)logical condition, such as residence time, inflow and/or run off, different hydrochemical compositions, like major ion concentrations (*Mezquita et al., 2001*; *Smith & Delorme, 2010*) that could have a undetected control on the species distribution. Salinity only indicates the ion concentration, but not the ion composition, a closer determination of the major ions could help to clarify the distribution pattern (*Smith & Horne, 2002*). Ion composition is a well-established factor determining the species composition and affecting species distribution (*Delorme, 2001*; *Mezquita et al., 2005*; *Pint et al., 2015*).

Also, other unmeasured (micro)habitat specific factors are possible, like substrate type, type and coverage of vegetation, food supply and flow energy (*Kiss, 2007*; *Marchegiano et al., 2017*; *Mezquita et al., 2005*).

However, not only habitat conditions, but also metacommunity dynamics (*e.g.*, inter-/intraspecific competition source–sink dynamics, dispersal rates, mass and rescue effects) are important drivers in distribution, abundance and life-cycle of species and can contract the structuring role of environmental parameters, especially in cosmopolitan species (*Guisan, Thuiller & Zimmermann, 2010*; *Leibold et al., 2004*). Species may migrate, for instance, to other microhabitats when the optimal niche is occupied, or competing species may develop contrary life-cycles to avoid competition (*Carbonel et al., 1988*).

This could imply that ostracod species considered within a small geographical scale are not predominantly controlled by the most commonly considered abiotic environmental parameters (*i.e.,* salinity, pH, temperature) and that they are probably not as euryoecious or generalistic as assumed so far.

The two species found exclusively in low saline (*P. marchica* and *N. monacha*) and higher saline (*P. compressa* and *I. gibba*) water bodies are not (except for *P. marchica*) abundant (<4%) and their occurrence is not significant enough to distinguish water bodies with respect to salinity. *Pseudocandona marchica* is more abundant but shows no (negative) correlation with salinity in CCA. Two species, *C. vidua* and P. *kraepelini*, occur in higher abundances in lower salinity water bodies (category a) and also grouped together in the cluster analysis. However, they also show no correlation with salinity or other parameters in the CCA. In water bodies with higher salinities (category b), *H. salina* and *C. torosa* are particularly abundant, and the CCA also shows a correlation between *H. salina* and salinity. In the cluster analysis, *H. salina* grouped together with *I. gibba*. However, *H. salina* shows higher abundances only in one water body (TE), and *I. gibba* also generally occurs only in two water bodies (TE2 and QW). Moreover, *C. vidua*, *P. kraepelini*, and *H. salina* also occur in other water bodies and show only slight differences in abundance in some

cases, especially in the taphocoenoses. This highlights the differences between bio- and taphocoenoses.

Considering correlations of the species abundances with the measured parameters (Fig. 9), it emerges that only few species correlate with specific parameters (*e.g.*, *L. inopinata* with temperature). However, the indifferent pattern of the species in the CCA indicates that species composition is water body-specific and not directly controlled by the measured parameters. Although the measured physico-chemical parameters salinity, pH, temperature and oxygen content cannot explain the species distribution, the parameters probably influence the population dynamics (*e.g.*, lifespan, temporal occurrence).

In the Mansfeld area, not only the occurrence of *C. torosa* is surprising, but also its distribution. *Cyprideis torosa* is biogeographically widespread, an ecologically opportunistic species and occurs in salinity ranges from freshwater to hypersaline (*De Deckker & Lord, 2017*). Several carapace characteristics (*e.g.*, sieve-pore shape, size, noding, valve outline) and shell geochemistry are linked to salinity (*Boomer, Frenzel & Feike, 2016*; *Frenzel, Ewald & Pint, 2017*; *Frenzel, Schulze & Pint, 2012*; *Grossi, Da Prato & Gliozzi, 2017*), therefore it has been utilized as index fossil to reconstruct palaeosalinity and -temperature (*e.g.*, (*Pint et al., 2012*; *Scharf, Herzog & Pint, 2017*)). Nearly all studies of living *C. torosa* are from coastal areas with brackish water conditions, *e.g.*, Portugal (*Cabral et al., 2017*), Spain (*Marco-Barba et al., 2012*; *Mezquita, Olmos & Oltra, 2000*), Belgium (*Heip, 1976*), and Germany (respectively, the North and Baltic Seas) (*Boomer, Frenzel & Feike, 2016*; *Frenzel, Ewald & Pint, 2017*; *Keyser & Aladin, 2004*; *Scharf, Herzog & Pint, 2017*). This suggests that *C. torosa* occurs in freshwater habitats and tolerate low salinity conditions, but seems to prefer brackish waters.

In the Mansfeld area, *C. torosa* occurs only between 0.9 to 7.9, having the highest population density at low saline water bodies (0.9 and 1.3). The position of *C. torosa* in the CCA near coordinate origin indicates that it is not affected by measured physico-chemical parameters and also in the cluster analyses *C. torosa* shows no similarities with other species. Further studies, ideally covering marginal marine and inland lakes are therefore necessary to prove whether *C. torosa* can be used as an indicator for salinity variability and -reconstructions since there could be autecological differences between marginal-marine and (low saline) inland populations.

*Wang et al. (2021)* figure out that populations from different regions are adapted to local aquatic environments and therefore develop specific preferences. Thus, the actual preference range of a species may be locally very narrow. The above-mentioned examples show, this assumption is not only restricted to a large spatial scale, but can also be valid on a local scale for spatially close water bodies with different conditions, such as the Mansfeld area. Therefore, each water body provides a specific combination of biotic and abiotic conditions for ostracod species. As a result, species seem to develop their specific population dynamics and/or different life-cycles, depending on the conditions they encounter (*Leibold et al., 2004*). Thus, no habitat specific species assemblages related to the documented physico-chemical parameters can be distinguished in this study.

## CONCLUSION

This actualistic-autecological survey focuses on the spatial and temporal distribution of ostracod assemblages in twelve saline inland water bodies with special emphasis to differences between bio- and taphocoenoses.

The study area represents a set of several water bodies within a relatively small geographical area. Affected by similar hydrological and climatological conditions, the investigated water bodies provide a salinity range of 0.7 to 20.7. Accordingly, ostracod species distribution was expected to predominantly reflect salinity gradients. However, analyses of species-environmental relationships not only revealed that salinity is not a major control on the distribution of species but also that there is no simple pattern in temporal-spatial ostracod species distribution. Thus, although most of the occurring species are considered as ecological generalists, species are not ubiquitous. Variations in physico-chemical parameters (temperature, conductivity, oxygen, and pH) did not help to explain temporal-spatial distribution of the ostracod species. Inferences regarding the species' life-cycles are elusive, as there are (strong) differences between the water bodies. Species may have developed water body specific life-cycles. Indices of postponed or bivoltine life-cycles and occurrences in slightly saline inland waters of *C. torosa* differs from observations from marginal marine habitats. Detailed studies are required to improve the use/potential as (palaeo-) salinity proxy and to verify that it is the same species, since this is the first record of living *C. torosa* in German inland waters, so far.

Furthermore, the relationship between abundance and species composition of living ostracods and related taphocoenoses is not straightforward. Strong differences between biocoenoses and taphocoenoses occurred even within water bodies and on very short time scales. This indicated that taphonomic processes can be very local including transport and relocation (*e.g.*, species loss, dispersal) and affect species assemblages even on short time scales (*i.e.,* monthly). This must be taken into account when fossil material is interpreted terms of biodiversity, (palaeo-) limnological and (palaeo-) ecological conditions. Although the dataset is complex and provides many information, further studies are required (with *e.g.*, shorter sampling interval, differentiation between juvenile stages) to capture all patterns lying below the species distribution. Our study provides insights about the complexity of ostracod species distribution in space and time. Therefore, in order to obtain reliable and conclusive data, ideally replicate sampling always should be carried out including a differentiation of living specimens and valves should be clearly indicated. This allows enhanced understanding of the spatial and temporal distribution ostracod species, and a clues of possible taphonomic processes. Future studies should include not only major ion composition of the ambient water, but also factors that have received little attention in the past such as vegetation type, composition or texture of substrate, nutrient input, hydrologic conditions, and community dynamics. This will help to increase the understanding of species' autecology and finally improve the indicator potential of these generalists.

## ACKNOWLEDGEMENTS

We thank Birgit Schneider and Sylvia Haeßner for their support at the laboratory. We are grateful to Renate Matzke-Karasz for her help identifying the species. Peter Frenzel is kindly acknowledged for his help with data analyses. David Horne, Peter Frenzel and one anonymous reviewer are thanked for their helpful remarks to improve the manuscript.

### Funding

Funded by the Open Access Publishing Fund of Leipzig University supported by the German Research Foundation within the program Open Access Publication Funding. The funders had no role in study design, data collection and analysis, decision to publish, or preparation of the manuscript.

### Grant Disclosures

The following grant information was disclosed by the authors:
The German Research Foundation within the program Open Access Publication Funding.

### Competing Interests

The authors declare there are no competing interests.

### Author Contributions

- Marlene Hoehle conceived and designed the experiments, performed the experiments, analyzed the data, prepared figures and/or tables, authored or reviewed drafts of the article, and approved the final draft.
- Claudia Wrozyna conceived and designed the experiments, analyzed the data, authored or reviewed drafts of the article, and approved the final draft.

### Data Availability

  The raw data are available in the Supplementary Files.

### Supplemental Information

Supplemental information for this article can be found online at http://dx.doi.org/10.7717/peerj.13668#supplemental-information.

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
