# Peer review of "Spatio-temporal distribution of ostracod species in saline inland lakes (Mansfeld lake area; Central Germany)"

_PeerJ, doi:10.7717/peerj.13668_

## Round 0.1 · original submission · Major Revisions

Three recognized experts have assessed your manuscript and identified a number of issues that the manuscript revision should resolve. In order to be accepted, the manuscript needs a comprehensive revision. Please consider the proposed changes as mandatory.

Please note also the annotated manuscripts provided by reviewers 2 and 3.

I look forward to your revised manuscript.

·

Excellent Review

This review has been rated excellent by staff (in the top 15% of reviews)
EDITOR COMMENT
The reviewer provided an extremely detailed, knowledgeable, and constructive response to the submitted manuscript. I was particularly impressed by their efforts to show the authors ways to use the data set even more optimally. In this way, they fulfilled the task of the scientific peer review process in a noteable and exemplary way: not only to identify weaknesses in submitted work, but also to make the best of a submitted manuscript in cooperation with the authors.

Basic reporting

The structure of the manuscript conforms to PeerJ standards. The introduction provides useful and appropriate context for the research; below I suggest the inclusion of more explanatory text about certain aspects, and clearer identification of research questions, aims and objectives. The text is written in good English, but it can and should be improved in the interests of clarity and completeness. I recommend that once revisions are completed the authors should seek the help of a native English-speaking scientist to revise and improve the English. Figures are well designed and informative but in some cases very complex and it takes time to understand what they are showing (more informative captions would help); some are of poor resolution and look fuzzy or pixellated when magnified sufficiently to study them closely (I am looking at the separate figure files, not just the figures in the compiled pdf). This issue affects Fig. 1 especially which includes some symbols and names that are practically illegible. Literature is well referenced and relevant; I have suggested (below, under General comments) some additional references that I think would be useful. Raw data are supplied as supplementary spreadsheets.

Experimental design

The submission presents original primary research that is within the scope of the journal. As noted above and in more detail below, research questions are more implicit than explicit and need to be defined more clearly; nevertheless, the research is certainly aimed at filling a significant research gap. The execution of the research appears rigorous and of a high technical standard. Some methods are not described in sufficient detail to facilitate replication, but this problem should be easy to rectify. See below (under general comments) for detailed criticisms and suggestions.

Validity of the findings

All underlying data have been provided as supplementary information; they appear to be robust and statistically sound; the acquisition of such a comprehensive dataset is impressive and commendable. However, in my opinion the analyses, interpretation and conclusions do not always do justice to the impressive data and I offer some suggestions (below) for ways of exploiting the data more fully. “Conclusions” could be better linked to original research questions (once these are made more explicit); some acknowledgement of limitations of the research design, methods and results could usefully be included in the “Conclusions”, together with suggestions of how future research design might be improved. In spite of criticisms, the overall research findings are significant and valuable, and I trust that the authors will not find it too onerous to address the detailed critical points and suggestions made below (under general Comments)..

Additional comments

This submission represents a valuable contribution to understanding the ecology and taphonomy of assemblages of ostracods, a crustacean group valuable for palaeoenvironmental reconstruction. The research is underpinned by extensive fieldwork and laboratory analyses. The longer I spent exploring the authors’ raw data, analyses and interpretation, the more I was impressed by what they offer, but at the same time I developed a strong feeling that the dataset was under-utilised and could be analysed and interpreted in greater detail. Although there are shortcomings in various aspects of the submission (and these are inevitably the focus of my detailed comments below) which should be acknowledged, I believe they can all be addressed positively in ways that will result in a more robust and useful publication.

Figures and tables

Figure 3, running over two pages, presents a lot of useful data and is very informative once you have worked out what it all means. It needs a fully explanatory caption, however; it took me some considerable study of the figure and relevant main text sections to work out that the small pie charts show % abundance in taphocoenoses and the coloured blocks show % abundance in the biocoenoses (working this out was not helped by the fact that some comments in the main text do not make it clear whether they apply to biocoenoses or taphocoenoses). The caption should explain all abbreviations; I could work out that a = adult and j = juvenile, but this should be made clear, and the abbreviations for Sample ID (OT2, OT etc.) should be explained or at least cross-referenced to a look-up table, e.g., the caption could state “Sample IDs as in Table 1”. It is actually quite confusing that the figures and tables refer to “Sample-ID” but the main text refers to “water bodies”, in both cases matched with the same abbreviations (OT2, OT etc.). The use of full stops and commas in numbers is confusing, as it shows total numbers of valves as 1.019, 1.452, 1.263 etc., which makes the full stops look like decimal points (these numbers should be 1,019, 1,452, 1,263 etc.). The salinity values on this figure, on the other hand, feature commas where decimal points are required: 0,7, 4,4 etc. should be 0.7, 4.4.... (as is done correctly in the main text). Practical salinity units actually have no units of measurement (confusing, I know) and should really be reported as simple numbers (perhaps with a statement in Materials and Methods to the effect that “all salinity values are in practical salinity units”), but the use of “psu” as an apparent unit of measurement is now so widespread it is probably not worth worrying about.

Figure 5 needs more explanation in the caption. In this case the caption states “Abbreviations of waterbodies as in Table 1”, but they are actually given as sample IDs in Table 1 (also in Fig. 3), although of course they match sampling site names, which are the names of waterbodies (but some waterbodies have more than one sampling site). More informative site/waterbody descriptions (especially in the “habitat type” column) would be helpful. For example, “lake” might be any of several different lacustrine habitats (littoral, sublittoral, profundal, mud, sand, stones, aquatic macrophytes etc.); is “Teufe 2”, described as “lake temporary”, an ephemeral/temporary lake or a marginal part of a permanent lake that dries out when lake level falls? Is “Quelle im Wald” a name or a description (e.g., “a spring in the woods”)? English versions of the names, as descriptors, could be helpful. The figure caption tells us that it shows “Ratios of adult and juvenile ostracod species in the water bodies”; I think it shows adult:juvenile ratios of ostracods and I would assume that the black part of each bar represents % adults and the grey part % juveniles (this needs to be clarified in the caption), but no species are named, so do these bars represent combined data from all species collected at each sampling site (i.e., these are adult:juvenile ratios of assemblages, not of individual species)? This is not really made any clearer in the relevant section of main text (lines 455-462), and it needs to be clarified.

Fig. 9 needs to be adjusted (redrawn if necessary) so that all letter codes are clearly visible and associated with, but not overlapping, the data points, lines or each other (some are so tightly clustered and overlapping as to be illegible). The figure also needs a clearer explanatory caption; it states “species code similar to Fig. 3” but they should be exactly the same as in Fig. 3 (e.g., “Cn”, not “CN” for Candona neglecta) so as to distinguish them more clearly from sample ID codes. Preferably different symbols should be used for the species and the waterbody/sample ID data points.


Materials and Methods

It is evident from “Materials and Methods” and “Results” that the study differentiated between live (with soft parts) and dead (empty valves and carapaces) ostracods; exactly how this was achieved is not very clearly explained (lines 145 -152), however. Were ostracods ever observed alive, or were they assumed to have been alive at the time of collection because their carapaces contained soft parts (appendages etc.)? That would be a perfectly reasonable assumption, but it needs to be stated. Line 145: “The samples were fixed with 70% Ethanol to preserve living specimens.” Was this done in the field at the time of collection, or after the samples were returned to the laboratory? If the latter, how long were samples kept, and in what conditions, before ethanol was added?

In “Materials and Methods” it is stated (lines 151-152) that “Empty carapaces and valves were sorted by juveniles and adults. Different juvenile stages were not distinguished.” Although it does not seem to be made explicit in “Materials and Methods”, references to, e.g., “...biocoenoses which are also characterized by very variable adult/juvenile ratios” (lines 460-461) and the data shown in Fig. 5, as well as the raw data, show that specimens collected alive were also sorted into adults and juveniles. Perhaps this could made clearer.

I also wonder about the way in which the adult:juvenile ratios reported here were calculated, if the specimens were picked separately from two different size fractions (retained on the 500 and 250 micron sieves; lines 147-148). As long as each sample’s size fraction was picked in its entirety then the counts could simply be added together, without need to consider proportional adjustments; I am fairly certain this was the case, but it needs to be stated clearly to inspire confidence in the data. Additional picking of ostracods from the >125 micron fraction might have yielded smaller juveniles of many species, although they would probably be difficult to identify to species level. In the case of D. stevensoni and Cyprideis torosa there is the added complication that these species have brood care, the eggs and the earliest hatched instars being retained in the brood chamber of the adult female (something that does not seem to be considered in the submission).


Salinity

It is stated in “Materials and Methods” that electrical conductivity was measured in situ in the waterbodies sampled (line 141), along with temperature, pH and oxygen concentration. In the “Results” section it is stated (line 184) that “Conductivity values range between 1.3 mS/cm (OT2) and 26.7 mS/cm (Fig 2 B)”. Elsewhere (e.g., lines 300-313) there are numerous references to salinity (values presented as practical salinity units), both for the waterbodies sampled (lines 304-305: “Water bodies were also classified in ascending order according to salinity values.”) and in terms of the salinity tolerances of the species recorded. The relationship between salinity and electrical conductivity should be explained; the two are not simply synonymous, and salinities reported in the literature may have been measured in different ways and therefore not always be directly comparable (especially true in non-marine waters).

In the “Conclusion” section there is a statement (lines 757-760) that “The water bodies show differences in ostracod species assemblage composition which cannot explained by measured ecological parameters. Especially, salinity has a comparably small influence on the ostracod species distribution.” Other possibly influential factors are mentioned (lines 765-766) such as “vegetation types, composition or texture of substrate, nutrient input, hydrologic conditions”, but the authors seem unaware that the solute chemistry of waterbodies is well established as a significant factor affecting species distributions. For example, different species have preferences for Calcium-, Sulphate- or Chloride-dominated waters (see Fig. 6 in Pint et al., 2015, already cited in this submission), or for bicarbonate-enriched or bicarbonate-depleted waters (see, e.g., Smith, A.J. & Horne, D. J. 2002 Ecology of marine, marginal marine and nonmarine ostracodes. In Holmes, J. A. & Chivas, A. R. (eds), The Ostracoda: Applications in Quaternary research, AGU Geophysical Monograph Series, Vol. 131, 37-64). To equate water chemistry to salinity (lines 611-612: “water chemistry (i.e., salinity)” is an over-simplification; the influence of solute chemistry needs to be acknowledged, at least.

The collection of abundant living Cyprideis torosa inhabiting inland waterbodies (especially some with very low salinity/conductivity) is of considerable importance and deserves greater prominence in the text; it could also be highlighted in the abstract and in “Conclusions”. Such apparently anomalous occurrences are quite frequently mentioned in the literature but as far as I am aware this is the first confirmation of living specimens in German inland waters. The discussion of these occurrences with respect to salinity (lines 718-728) is useful but it could be expanded and considered more critically. Cyprideis torosa has been studied extensively and is far more than just “an indicator species for brackish water”, since variable morphological features (carapace size, sieve pore shape, phenotypic noding) and shell geochemistry (stable isotopes, trace elements) have been calibrated for use as proxies for palaeosalinity. See the special 2017 issue of Journal of Micropalaeontology devoted to papers on Cyprideis torosa, starting with: De Deckker, P. & Lord, A.R. 2017. Cyprideis torosa: a model organism for the Ostracoda? Journal of Micropalaeontology 36, 3-6.


Biocoenoses and taphocoenoses

I very much agree with the authors’ view that ostracod research has often differentiated insufficiently between living ostracods (biocoenoses) and their empty valves (taphocoenoses) (lines 102-103). However, these terms should be explained, not only to ensure that the reader understands them, but also to demonstrate what the authors understand by them, because they have often been applied incorrectly in ostracod studies (this was explained by Boomer, Horne & Slipper, 2003, a reference cited only in a more general context in this submission). Assemblages of ostracods represented by empty carapaces and/or valves (resulting from moulting or death) may often be taphocoenoses (transported, sorted and mixed before final deposition) but there are other possibilities (for example in a very small waterbody with little or no potential for transport, sorting and mixing, an autochthonous thanatocoenosis may accumulate). Many natural assemblages probably comprise both thanatocoenosis and taphocoenosis components. Even after burial of an assemblage there may be taphonomic processes such as diagenesis to take into account. Pint & Frenzel (2017: p. 118) (a reference already cited in this submission) have understood these issues well: “Analyses on fossil ostracod associations rely mainly on taphocoenoses. Using preservation and population structure as indicators, allochthonous elements may be identified... but this criterion is hard to use for well-preserved specimens of rare species. Hence, some portions of the studied associations have to be accepted as unidentified allochthonous elements. This problem is especially serious in estuaries, where inflowing water from the mainland and also the sea carries ostracod shells to the sites. Similarly, athalassic saline lakes and coastal lagoons with abundant C. torosa may be affected by transported limnic taxa from rivers or marine taxa from the sea. The produced taphocoenosis is, of course, not completely an assemblage of animals living together or in the same location over the seasons but a mixture of autochthonous and allochthonous elements. This makes palaeoecological interpretation more complicated but tells us something about provenance and, hence, the character and interconnectedness of the studied water body. We confess that we are not able to judge the autochthonous character of every individual registered in our count, but use its presence for the characterization of the waters inhabited by C. torosa, thus integrating ecological and taphonomical information”. I welcome the present authors’ apparent intention of addressing such issues by studying ostracods in the “natural laboratory” of the Mansfeld lakes area waterbodies. As submitted, however, these intentions are mostly only implicit; the only exception seems to be the final sentence of the Introduction (lines 120-122): “By repeated sampling, we also wanted to investigate if there is a seasonal effect on the distribution and abundance of the ostracod species related to the specific life-cycle of the species”. Knowledge gaps are identified with reference to relevant literature, but the authors need to clarify how the research was designed to fill those gaps; the purpose of the research really needs to be made more explicit by means of clearly stated research questions and aims. That being done, it would then be helpful to refer to aims and objectives when explaining the methods; explain not only the method but also its purpose (e.g., “In order to determine seasonal variations in ostracod species occurrences and environmental factors, three sampling campaigns (April, June, September 2019) were carried out in the Mansfeld lake area”). A limitation of the research is that juvenile valves were not differentiated according to moult stages; doing so could have been very informative in assessing which components of each assemblage had been most subject to transport and sorting (again, see Boomer, Horne & Slipper, 2003); I appreciate that time constraints mean that you cannot do everything, but it would be helpful if the authors themselves would acknowledge such a limitation rather than leaving it for readers to recognise and criticise.

Lines 597-598: “An indication of valve transportation is the extremely high number of adult valves compared to juvenile valves in most water bodies”. I agree, but this is an observation that could usefully be explained, in terms of processes, in more detail. Make it clear that this is an indication of differential transport (removal by winnowing, probably) of juvenile valves, leaving behind the larger/heavier adult valves. In an enclosed environment (e.g., an aquarium) juvenile valves should greatly exceed adults in number, because ostracods typically moult eight times before reaching maturity and thus each adult individual has already contributed sixteen juvenile valves to the sediment. Transport could be by currents or wave action, for example. Wave action around the shallow margins of a lake will most easily resuspend the smaller juvenile valves which are then redeposited in deeper water; assemblages in shallow marginal waters are thus dominated by adults and depleted in juveniles. It is a pity that juveniles were not differentiated and assigned to different juvenile stages, because such data could be very informative about transport and size-sorting (see, e.g., Boomer et al. 2003, already cited in this submission). Nevertheless, the simple counts of adults and undifferentiated juveniles in the raw data could be used more effectively, in combination with published information about the life-cycles of species.


Life-cycles

Although life-cycles of ostracod species are quite frequently mentioned in the “Introduction” and “Discussion” sections, I could find few indications of any real consideration of the known life-cycles and seasonality of any named species in the context of the research reported here (not even under “Inferences on species life-cycle”; lines 473-492) The only exceptions are perhaps the mention of Darwinula stevensoni having a life-span of <1 to 4 years (line 710) (but no mention of the fact that this species has brood care), and salinity-related differences in life cycles of Limnocythere inopinata (lines 711-713) (citing Geiger, 1998, which is missing from the reference list). The latter most likely comes from Geiger et al. 1998, who found that, in L. inopinata, parthenogenetic clonal lineages respectively adapted to saline water and to fresh water never co-occur, and the saline water populations overwinter as adults and juveniles while the freshwater ones disappear in winter; this implies that freshwater populations overwinter as eggs. The present authors’ data is uninformative in this respect since they did not sample in the winter months. It is worth noting that L. inopinata is a Holarctic species that also has sexual populations, for example in North America where the parthenogenetic and sexual populations have different though overlapping distributions that are probably related to water chemistry.

Therefore, despite the concluding comment of the Introduction (lines 120-122) that “By repeated sampling, we also wanted to investigate if there is a seasonal effect on the distribution and abundance of the ostracod species related to the specific life-cycle of the species”, it seems that this aspirational investigation was not pursued to any significant extent. There could be many good reasons why this aspect of the intended research could not be addressed; at least some brief explanation of why it could not be achieved should be included in discussion. Alternatively, if the authors would like to pursue it and take into account what is known about the life-cycles of the species they identified, Claude Meisch’s (2000) monograph (which they have already cited a few times) would be a good starting point as it includes summary life histories of many species, for example Candona neglecta (a permanent form with two generations per year), Sarscypridopsis aculeata (two or three generations per year, overwintering mainly as eggs) and Cyprideis torosa (with brood care; one generation per year, overwintering as adults and juveniles).

I wondered: were juveniles as well as adults in the biocoenoses and taphocoenoses identified to species level? I looked at the Supplementary spreadsheets and found that they were. For example, the April TS biocoenosis has Darwinula stevensoni with 48 adults and 33 juveniles, April TE biocoenosis has Cyprideis torosa with 68 adults and 46 juveniles. These impressive datasets seem to have been under-utilised, with only a few comments in the main text, such as (lines 481-483): “Almost all species show the same pattern between adults and juveniles in the biocoenoses. For example, adult D. stevensoni are most abundant in summer, juveniles as well”. It would be useful to at least make greater use of the graphic presentation of seasonally changing adult:juvenile ratios of selected species (summarized for all waterbodies) shown in Fig.7, but ideally to also consider data for selected species and selected waterbodies, and to attempt to match these with known life-cycles. I think this would be quite straightforward to do, and not particularly time-consuming. For example: a;j ratios of Heterocypris salina at TE (lake, mud substrate) were 2:61 in April, 12:50 in June and 1:5 in September, and at BS (lake, sand substrate) they were 5:1 in April, 13:12 in June and 13:14 in September. This species can produce two or three generations per year, taking only 45 days to complete a generation’s life-cycle, and overwinters as eggs (Meisch, 2000). I suggest, tentatively that the dominance of juveniles in the April and June collections might represent two successive generations. The equivalent taphocoenoses data seem to support this idea, and it is interesting to note the significant numbers of juvenile carapaces recorded, because these imply juvenile mortality (moulted carapaces are typically disarticulated into separate valves) which could be indicative of an adverse perturbation of their environment. The lower numbers at BS compared to TE (in both the biocoenoses and the taphocoenoses) might be explained by the different substrates sampled at these two sites, and it is tempting to infer that the BS assemblages (from sand, a higher-energy substrate than mud) have been more affected by transport. I may be over-interpreting the data, and the authors should do their own analyses; this example is simply intended to demonstrate that much more could be done with the available raw data.

ref.: Geiger, W., Otero, M., Rossi, V. 1998. Clonal ecological diversity. In: Martens, K. (ed.) Sex and parthenogenesis. Evolutionary ecology of reproductive modes in non-marine ostracods, 243-256. Bakhuys Publishers.


Conclusions

It is apparent from the “Discussion” section that the multitude of potentially influential variables makes it difficult to determine the factors that may drive the differences between biocoenoses and taphocoenoses. In “Conclusions”, some attention might usefully be paid to lessons learned from the present study and suggestions for improving the design of future field collecting campaigns in ways aimed at reducing the number of variables (for example, seeking out and sampling the same kinds of substrate or habitat in each waterbody).


Other comments

lines 251-252:
Family: Notodromatidae Kaufmann, 1900
Subfamily: Notodromatinae Kaufmann, 1900

Correct spellings are Notodromadidae and Notodromadinae

Please check throughout for spelling errors, especially in taxonomic names incorrect endings of names are commonly introduced by autocorrecting software). For example, in the caption for Fig. 3 “Potamocypris arcuate” should be “Potamocypris arcuata” and the same error is quite frequent in the main text. Similarly “S. aculeate” should be “S. aculeata” (line 393). Fig. 4 caption: “thaphocoenoses” should be “taphocoenoses”.

David J. Horne 15th February 2022

·

Basic reporting

Despite the English text is of high quality there is still room for improvements. I indicated mispellings, stylistic problems etc. within the annotated version of the manuscript. I have to stress, however, that I am not a native speaker, so check carefully. The use of the term abundance is sometimes confusing because it is not always clear for me if abundance, i.e. the number of individuals in a given volume/area, or relative abundance [%] is addressed.
The literature used gives a good background for the discussion of data obtained. An exception is the largely missing description of the study area. I suggest to provide a chapter Study Area using some information from Introduction and M & M and adding data about variability of salinity and maybe pollution. Some of the water bodies investigated are monitored, so information could be gathered. The variability, especially in salinity, is crucial for the interpretation of associations from taphocoenoses. The lack of this type of information is a shortcoming of the manuscript under review.
The results chapter and, in a lesser degree, the discussion could be better structured enabling the reader to follow more easily. This is not an easy task with the wealth of data obtained and ambiguous patterns of distribution. I know this from my own painful experiences with similar datasets. Maybe focusing and a little shortening of the text could help. Furthermore, as suggested within the annotated manuscript, the authors could add a table with their classification of sites as reference for the reader. Clear research questions and focusing discussion to them could also help to foster a clear structure.
The diagrams and other figures are mostly well prepared, but their captions are often too short. Details are given within the annotated manuscript. The diagrams of Fig. 7 are too complex and should be split.

Experimental design

The study clearly adds to our understanding of ostracod associations from athalassic saline water bodies, a field not well investigated so far, and compares living assemblages with taphocoenoses what is crucial for applying Recent data to palaeoenvironmental reconstructions. I highly appreciate this. I feel, however, the authors could point out the relevance of their paper more clearly by focusing on selected research questions.
There is an important information missing but needed for interpreting distribution patterns: The area/volume of sampling is not given. Based on the text I conclude that not all samples have the same size, so it would need standardisation of abundance data to compare abundance. If abundance is standardised within the manuscript, it should be mentioned and explained how.
Using the adult/juvenile ratio is important not only for taphonomic processes but also for evaluating seasonal population dynamics as shown by the authors. Please, be always aware that this ratio is highly taxon-specific in your size fraction >250 µm.

Validity of the findings

The first finding of living Cyprideis torosa in athalassic waters of Central Europe shoud be stressed as highlight. Many researchers were hunting for such material before but without success. This is mentioned but not recognisable as a remarkable finding.
The living/adult ratio was calculated but not used for taphonomic interpretation. This should be added, a diagram plotting the sites with standardised abundances in both groups could be helpful by displaying productivity and preservation potential of all sites.
The conclusions are too negative in implications for palaeoenvironmental analyses by my opinion. The data set is rather large in terms of efforts but probably too small and hererogenous to discover all patterns lying below. For instance, the statement of salinity not being a driving factor of distribution of species is based on the missing of the brackish water taxon Cyprideis torosa at highest salinities, but this is not diminishing its worth as brackish water indicator because it is an opportunistic holeuryhaline species. I am convinced, the heterogenous distribution pattern of other species is not hampering their use for salinity reconstructions, what could be proven with applying the salinity tarnsfer function used by Pint et al. (2015) to taphocoenoses of the present study.
The seasonal occurence is not discussed referring to literature (e.g. Meisch 2000).
Habitat descriptions exist but are not used for discussion of distribution patterns, at least not recognisable for me.

Additional comments

See scan of annotated manuscript for more details.

Reviewer 3 ·

Basic reporting

The manuscript titled “Spatio-temporal distribution of ostracod species in saline inland lakes (Mansfeld lake area; Central Germany)” submitted by the authors Marlene Hoehle and Claudia Wrozyna is written in clear, scientifically sound manner. The abstract is concise and text is well organized within required units of IMRAD structure. Appropriate taxonomical nomenclature is used throughout the text and the technical and scientific quality of the illustrations is acceptable. Apart from few corrections suggested, the literature cited is adequate and it is appropriately listed. The quality of the English language is satisfactory. Comments on the references and some technical details in particular parts of the text and on figures are given in the additional PDF.

Experimental design

The manuscript contains valuable and new information on the subject of ostracod ecological preferences and dynamics of biocenoses and taphocenoses, with a critical analysis important for the implications in palaeoenvironmental reconstruction. The experiment and research questions are well set, and appropriate methods have been used to obtain and accurately analyze the data.

Validity of the findings

The conclusions drawn from this research are well stated and supported by the results, and adequately corroborated with literature citations.

Additional comments

A minor revision is needed and the authors are asked to take in consideration the more detailed comments marked in the particular parts of the text, and given in the annotated PDF of the manuscript draft.

Annotated reviews are not available for download in order to protect the identity of reviewers who chose to remain anonymous.

---

## Round 0.2 · Minor Revisions

Thank you very much for the thorough and far-reaching revision. Now, there are only some minor (and mostly editorial) points left which you should consider in a final revision, based on the comments in the annotated manuscript provided by the reviewer.

Reviewer 3 ·

Basic reporting

The article is written in comprehensive way, with professional scientific style. All required sections are included from Introduction to Conclusions, and the Abstract is in coherence with the manuscript content. Illustrations are well labelled and referred in the text.

Experimental design

This work presents an original and valuable study on the complex relationships of ostracods with their environment, tackling an important topic of taphocoenoses that are the only available material in paleoecology.
The description of methods is significantly improved comparing to the previous version.

Validity of the findings

The data was processed with a lot of care in the previous version already. Now, the considerable improvements in the Material and methods sections and the results enabled bettering the explanations in the discussion and conclusions. The new literature sources chosen to refer in the discussion are well selected and connected to the topic. However, there are some remarks to consider in the particular comments given in the additional file.

Additional comments

Additional, mostly technical details that need to be corrected are given in the PDF.

Annotated reviews are not available for download in order to protect the identity of reviewers who chose to remain anonymous.

---

## Round 0.3 · accepted · Accept

Thank you very much for the final revision. I look forward to see the manuscript published in due course.